


# Lateral transports of mass, inorganic nutrients and dissolved oxygen in the Cape Verde Frontal Zone in summer 2017

Nadia Burgoa[1], Francisco Machín[1], Ángel Rodríguez-Santana[1], Ángeles Marrero-Díaz[1], Xosé Antón Álvarez-Salgado[2], María Dolores Gelado-Caballero[3], and Javier Arístegui[4]

[1]Departamento de Física, Universidad de Las Palmas de Gran Canaria, Spain
[2]CSIC Instituto de Investigacións Mariñas, Vigo, Spain
[3]Departamento de Química, Universidad de Las Palmas de Gran Canaria, Spain
[4]Instituto de Oceanografía y Cambio Global, Universidad de Las Palmas de Gran Canaria, Spain

**Correspondence:** Nadia Burgoa (nadia.burgoa@ulpgc.es)

**Abstract.** The circulation patterns in the confluence of the North Atlantic Subtropical and Tropical gyres delimited by the Cape Verde Frontal Zone (CVFZ) in summer 2017 were examined. Hydrology, dissolved oxygen ($O_2$) and inorganic nutrients data collected in a closed box embracing the CVFZ allowed estimating transports of water masses, $O_2$ and inorganic nutrients for the first time. Higher transports occurred mainly at the surface and central waters, and were moderately affected by the

5 Cape Verde Front located in the southeastern part of the domain. Thus, the front conditioned the meridional transports, acting as a barrier between North and South Atlantic Central waters. Specifically, $-3.2 \pm 1.7$ Sv entered through north and east and $6.7 \pm 1.7$ Sv left through west and south transects. At intermediate levels, the most important source came from the south with $-2.2 \pm 1.5$ Sv of modified Antarctic Intermediate water, moderately affected by the circulation pattern above. The transports of $O_2$ and inorganic nutrients conditioned by their distributions behaved quite similar to mass transports. The most intense and

10 important transports of $O_2$ and inorganic nutrients occurred in the deepest layer of central waters and in the shallowest two layers of intermediate waters where inorganic nutrients accumulated and large differences in concentrations of $O_2$ were found, especially in the deepest layer of central waters between the northeast and southeast zones. In these three layers, transports of $O_2$ and inorganic nutrients came from the east and south and they left northward and westward. This circulation pattern delivers inorganic nutrient from east and south to oligotrophic waters of north and west of CVFZ. Nevertheless, it can also

hinders the ventilation of the deepest layer of central waters and the shallowest two layers of intermediate waters.

## 1 INTRODUCTION

The Cape Verde Basin (CVB) is located in the eastern boundary of the North Atlantic Ocean where the subtropical region meets the tropical one. This area is influenced by the south-eastern extension of the North Atlantic subtropical gyre, NASG (Stramma and Siedler, 1988), the north-eastern extension of the North Atlantic tropical gyre, NATG (Siedler et al., 1992),

and the upwelling region off NW Africa (Ekman, 1923; Tomczak, 1979; Hughes and Barton, 1974; Hempel, 1982). Inside this domain, the Cape Verde Frontal Zone (CVFZ) extents from Cape Blanc to Cape Verde Islands as a northeast-southwest frontier between subtropical and tropical waters (Zenk et al., 1991). In addition, the coastal upwelling front (CUF) along the





Mauritanian coast, until Cape Blanc/Cape Verde in summer/winter, is another barrier that separates stratified inner-gyre waters from more homogeneous slope waters in the CVB. These two frontal systems are also sources of mesoscalar and submesoscalar variability (Capet et al., 2008; Thomas, 2008). Interleaving mixing processes related with the thermohaline horizontal gradients in the Cape Verde Front (CVF) and filaments associated to the CUF have also been documented (Pérez-Rodríguez et al., 2001;

Martínez-Marrero et al., 2008; Meunier et al., 2012; Hosegood et al., 2017).

The northern side of the CVFZ is mainly occupied by North Atlantic Central Water (NACW), which flows southward as the Canary Current (CC). Once it reaches the CVFZ, it turns offshore as the North Equatorial Current (NEC) (Stramma, 1984), giving rise to a shadow zone of poorly ventilated waters (Luyten et al., 1983). Additionally, long-lived eddies generated downstream of the Canary Islands (Sangrà et al., 2009; Barceló-Llull et al., 2017) can reach to CVFZ and significantly contribute to

the westward circulation (Sangrà et al., 2009). Between the Canary Islands and Cape Blanc, the trade winds are intense enough throughout the year to force a permanent upwelling (Benazzouz et al., 2014), which in turn generates an intense southward coastal jet, the Canary Upwelling Current (CUC) (Pelegrí et al., 2005, 2006). Below the CUC and over the continental slope, the Poleward Undercurrent (PUC) flows northward intensely (Barton, 1989; Machín and Pelegrí, 2009; Machín et al., 2010).

South Atlantic Central Water (SACW) is the main water mass at the southern side of the CVFZ. This water mass is formed

at the subtropical South Atlantic and it modifies after crossing the tropical regions (Peña-Izquierdo et al., 2015). These tropical waters move anticlockwise around the Guinea Dome (GD) with a seasonal variability mainly driven by latitudinal changes in the Inter-Tropical Converge Zone (ITCZ) (Siedler et al., 1992). The circulation of the GD is composed by the eastward North Equatorial Counter Current (NECC), which feeds the northward transport of SACW via Mauritanian Current (MC) and PUC. In summer, GD intensifies as a result of the northward penetration of ITCZ (Castellanos et al., 2015). In addition, the northward

flow along the African coast intensifies due to the relaxation of trade winds at latitudes south of Cape Blanc, so MC and PUC can reach just south of Cape Blanc in this season (Siedler et al., 1992; Lázaro et al., 2005).

The encountering of southward-flowing CC and CUC with northward-flowing PUC and MC leads to a confluence at the CVFZ which fosters the export of water offshore, with its maximum values in summer (Pastor et al., 2008). The subtropical and tropical waters exported along the CVFZ have different properties and characteristics. NACW is a relatively young, salty

and warm water mass with nutrient-poor and oxygen-rich concentrations. SACW is an older water mass fresher and colder than NACW, modified while travelling through tropical regions; hence, SACW that reaches the CVFZ is a nutrient-rich and oxygen-poor water mass (Tomczak, 1981; Zenk et al., 1991; Pastor et al., 2008; Martínez-Marrero et al., 2008; Pastor et al., 2012; Peña-Izquierdo et al., 2015). When nutrient-rich SACW reaches the CVFZ, the front drives SACW into the southeastern edge of the nutrient-poor NASG, a process that helps to maintain a high primary production far from this coast as

seen in the giant filament off Cape Blanc (Gabric et al., 1993; Pastor et al., 2013). North of Cape Blanc, the developed upwelling is a second source of nutrient-rich waters which are transported offshore as upwelling filaments and mesoscale eddies (García-Muñoz et al., 2005; Álvarez-Salgado et al., 2007; Ruiz et al., 2014; Lovecchio et al., 2017, 2018).

On the other hand, the latitudinal propagation of modified Antarctic Intermediate Water (AAIW) along the eastern margin of both the NASG and NATG has been previously documented (Machín et al., 2006; Machín and Pelegrí, 2009; Machín et al.,





2010). AAIW is a relative fresh and cold water mass with high inorganic nutrient concentration and low dissolved oxygen concentration, which at this latitude follows the African slope northward at 700-900 m depth.

Below the euphotic layer, the distribution of $O_2$ and inorganic nutrients are determined by both biogeochemical and physical processes (Pelegrí and Benazzouz, 2015). The main biogeochemical processes are related with the availability of organic

matter, dissolved oxygen and inorganic nutrients at the source regions and also with the remineralization processes; on the other side, the main physical processes are associated with both the vertical connection between surface and subsurface waters and also with the circulation patterns at subsurface waters (Peña-Izquierdo et al., 2015; Pelegrí and Benazzouz, 2015). As a consequence, the $O_2$ and inorganic nutrients concentrations may vary depending on the interplay between the local rate of remineralization and the rate of water supply (Pelegrí and Benazzouz, 2015). In other words, the dynamic differences between the

subtropical and tropical regions separated by the CVFZ, and also between the interior-gyre and the upwelling region separated by the CUF, establish distinct biogeochemical domains with substantial differences in $O_2$ and inorganic nutrient concentrations at the CVB.

In recent years, several authors have addressed the distribution of $O_2$ and inorganic nutrients and also their link to the physical processes taking into account the water mass distributions and transports in this margin of the North Atlantic Ocean

(Pelegrí et al., 2006; Machín et al., 2006; Pastor et al., 2008; Álvarez and Álvarez-Salgado, 2009; Peña-Izquierdo et al., 2012; Pastor et al., 2013; Peña-Izquierdo et al., 2015; Hosegood et al., 2017; Burgoa et al., 2020). These physical processes range from the regional scale, usually associated to advective processes, to mesoscale and smaller scales, usually associated to mixing processes (Pastor et al., 2008). These works consider remineralization as the main biochemical process inside NASG and NATG.

This manuscript aims to address the circulation patterns and the physical processes behind the distribution of $O_2$ and inorganic nutrients at the CVFZ, a domain where *in situ* data availability has historically been very limited. Secondly, the location and intensity of the CVF in summer is assessed, a front acting both a barrier and a source of meso- and submesoscale variability, having influence on the transports of mass, $O_2$ and inorganic nutrients. The rest of the manuscript is organized in three main sections. Section 2 presents the methodology, where the sampling performed during the FLUXES-I cruise and the modelling

description are given. Then, section 3 presents results in terms of the hydrography and water masses, absolute velocity field, and transports of mass, dissolved oxygen and inorganic nutrients. The manuscript is closed with a discussion of the results and the main conclusions.

## 2    METHODOLOGY

### 2.1    The cruise

FLUXES-I cruise was carried out on summer 2017, from July 14th to August 8th aboard the R/V Sarmiento de Gamboa as part of the research project FLUXES (Carbon Fluxes in a Coastal Upwelling System – Cape Blanc, NW Africa). The 35 sampling stations were selected to form a closed box (pink dots in Fig. A1). The rosette sampler was a SBE 38 equipped with 24 Niskin bottles of 12 L. Conductivity-temperature-depth data (CTD) were collected with a SBE 911+ from the sea surface down to





more than 2000 m depth with a vertical resolution of 1 dbar. Stations 1, in the northeast corner, and 29, in the southeast corner, were very shallow, so both were discarded from the analyzes. The average distance between neighboring CTD stations was some 84 km. In addition, 39 expendable bathythermograph probes (XBT, T5 by Sippican) were deployed between most CTD stations (blue dots in Fig. A1). In all cases, the XBTs reached down to 2000 m thanks to a specific setup in the WinMK21

acquisition program and to a reduced boat speed during deployment (5 kn). Some XBTs were also removed (12, 19, 30, 31, 38, 39 and 40) due to recording errors. The northern transect (N) spanned from station 2 to 12 at $23°$N; the western one (W) was located at $26°$W from station 12 to 19; the southern zonal transect (S) at $17.5°$N ran from station 19 to 28, while the eastern transect (E) closed the box approximately at $18.57°$W from station 28 to 3.

Practical salinity, $S_P$ (UNESCO, 1985) was calibrated by analyzing 51 water samples with a Portasal model 8410A sali-
10 nometer with an accuracy and precision within the values recommended by WOCE. An oxygen sensor SBE43 was interfaced with the CTD system during the cruise. In total, 417 samples were collected to calibrate the dissolved oxygen sensor on the rossette. The final precision obtained was $\pm 0.53$ $\mu$mol kg$^{-1}$. The dissolved inorganic nutrients analyzed in this work were nitrates (NO$_3$), phosphates (PO$_4$) and silicates (SiO$_4$H$_4$). 419 water samples were collected by the Niskin bottles to 25 mL polyethylene bottles in all the stations and depths. The samples were frozen at -20°C until their analyzes in the base labora-
15 tory. NO$_3$, PO$_4$ and SiO$_4$H$_4$ concentrations were determined using a segmented flow Alliance Futura analyzer following the colorimetric methods proposed by Grasshoff et al. (1999).

### 2.2 Remote sensing and climatological datasets

ASCAT on Metop L4 daily global wind field (Bentamy and Fillon, 2012) with a spatial resolution of $0.25°$ made available by CERSAT (Centre ERS d' Archivage et de Traitement; http://cersat.ifremer.fr) was employed as the source of wind data to
20 estimate the Ekman transport. Freshwater flux was estimated from the average rates of evaporation and precipitation extracted from the Weather Research and Forecasting model (WRF (Powers et al., 2017)), provided with a spatial resolution of $0.125°$ and a temporal resolution of 12 h.

The climatological mean depths of the neutral density field during the summer season were calculated from the climatological temperature and salinity extracted from the World Ocean Atlas 2018 (Locarnini et al., 2018; Zweng et al., 2018, WOA18).
Moreover, WOA18 was also employed to generate a climatological neutral density field during the summer season to estimate a climatological geostrophic velocity field. On the other hand, 4 new stations in the N transect and 12 new stations in the S transect were produced with the WOA18 summer field with the aim to extend the N and S transects of the inverse model to the African coast.

SEALEVEL_GLO_PHY_L4_REP_OBSERVATIONS_008_047 product issued by Copernicus Marine Environment Mon-
30 itoring Service, CMEMS (http://marine.copernicus.eu) provided Level-4 Sea Surface Height (SSH) and derived variables as surface geostrophic currents, measured by multi-satellite altimetry observations over the global ocean with a spacial resolution of $0.25°$. These data capture the mesoscale structures and are helpful to validate the near-surface geostrophic field.

GLORYS 12V1 (GLOBAL_REANALYSIS_PHY_001_030 product also issued by CMEMS was used to diagnose the average dynamics during 2017 with a horizontal resolution of $1/12°$ at 50 standard depths.



All data treatments (*in situ*, operational and modelling), interpolations with Data-Interpolating Variational Analysis (DIVA, (Troupin et al., 2012)), the analyzes, the graphical representations, and the inverse model were coded in MATLAB (MATLAB, 2019). Additionally, the Smith–Sandwell bathymetry V19.1 (Smith and Sandwell, 1997) was used in all maps and full-depth vertical sections.

## 2.3  Improving resolution and derived variables

The XBTs deployed between CTD stations doubled the *in situ* temperature (*t*) horizontal resolution. A unified *in situ* temperature dataset from surface to 2000 m depth was constructed with the CTD and XBT profiles. The interpolation of *t* was optimally generated in each vertical section with DIVA once the horizontal and vertical correlation lengths, $L_x$ and $L_y$, and also the signal to noise ratio, $\lambda$, were defined for all transects. The vertical correlation length was $L_y = 50$ m, the horizontal correlation length was within the range $L_x = 110 - 120$ km and $\lambda = 4$. Once the interpolated *t* was validated by a comparison with the original XBT profiles, the same $L_x$, $L_y$, and $\lambda$ values were employed for the remaining hydrological and biochemical variables, i.e., $S_P$, $O_2$, $NO_3$, $PO_4$ and $SiO_4H_4$.

After these interpolations, $S_P$ was converted into Absolute Salinity ($S_A$, McDougall et al. (2012)), while *t* was converted into Conservative Temperature ($CT$, McDougall and Barker (2011)) as TEOS10 variables (IOC et al., 2010). In addition, neutral density ($\gamma_n$, Jackett and McDougall (1997)) was used as the density variable.

## 2.4  Velocity field and transports

The geostrophic velocity was calculated along the perimeter of the box referenced to $\gamma_n = 27.962 \, \mathrm{kg \, m^{-3}}$, where the geostrophic velocity was initially considered to be null. This isoneutral surface was the deepest common isoneutral for all the stations and it is located at around 2000 m depth. An inverse box model (Wunsch, 1978) was then applied to estimate the reference level velocities at the box boundaries. This inversion method provides the reference level geostrophic velocity field to estimate the absolute water mass transport through each transect of the closed box, a method that has been widely applied in the Atlantic Ocean (Martel and Wunsch, 1993; Paillet and Mercier, 1997; Ganachaud, 2003a; Machín et al., 2006; Pérez-Hernández et al., 2013; Hernández-Guerra et al., 2017; Fu et al., 2018).

Instead of working with the whole box, it was decided to close the box at the coast to avoid any issues in the temporal evolution of structures when closing the eastern transect with the northern one. To do so, WOA18 climatological nodes were used extending eastward the transects N and S. Therefore, the geostrophic velocities at the reference level were obtained from the inversion for transects N, W and S, while for the E transect those velocities were actually the annual mean extracted from GLORYS and interpolated to the reference level $\gamma_n = 27.962 \, \mathrm{kg \, m^{-3}}$ (obtained from WOA18).

The absolute velocity field allows transport estimates of $O_2$ and inorganic nutrients for the whole box. Those transports are obtained by multiplying their concentration times mass transports, so their concentrations are indeed interpolated to the positions where the geostrophic velocities were estimated. The uncertainties of mass transports estimations per layer and per water levels for transects N, W and S were extracted from the inverse model thanks to the *a priori* information compilation. On the other hand, the uncertainties of transect E for the mass transports were calculated with the velocity variance estimated





from the annual mean velocity of GLORYS. The uncertainties of transport estimates of $O_2$ and inorganic nutrients are relative to those of mass transport.

### 2.4.1 Characteristics and constrains of the inversion model

The inverse model is consistent with both geostrophy and conservation of mass, salt and heat, allowing an adjustment of Ekman

transports and freshwater flux. The model is made up of 8 layers divided by the free surface and 8 isoneutrals (26.46, 26.85, 27.162, 27.40, 27.62, 27.82, 27.922 and 27.962 $\mathrm{kg\,m^{-3}}$), taken essentially from those defined by Ganachaud (2003a) for the North Atlantic Ocean (Fig. A2). The inverse model takes into account mass conservation and salinity anomaly conservation per layer and also over the whole water column (Ganachaud, 2003b). Heat anomaly is introduced only in the deepest layer where it is also considered conservative. Salinity and heat are added as anomalies to improve the conditioning of the model

and reduce the linear dependency between equations (Ganachaud, 2003b). Therefore, the inverse model is composed of 19 equations (9 for mass conservation, 9 for salt anomaly conservation and 1 for heat anomaly conservation). Those equations are solved using a Gauss-Markov estimator for 69 unknowns, comprised of 65 reference level velocities, 3 unknowns for the Ekman transport adjustments (one per transect), and 1 unknown for the freshwater flux. In addition, it is necessary to provide *a priori* the uncertainties related to the noise of the equations ($R_{nn}$) and the unknowns ($R_{xx}$) in order to solve this undetermined

system composed of 19 equations and 69 unknowns.

The noise of each equation is dependent on the layer thickness, on the density field, and on the uncertainties of the unknowns (Ganachaud, 1999, 2003b; Machín et al., 2006). Thus, an analysis of the annual variability of the velocity is performed in the mean depths of the 8 layers. The velocity variance of each layer is estimated from the annual mean velocity provided by GLORYS. These variances are transformed into transport values multiplying by density and the vertical area of the section

involved. The uncertainty assigned to the total mass conservation equation is the sum of the uncertainties from the rest of the 8 mass conservation equations. The equations for salt and heat anomaly conservation depend on both the uncertainty of the mass transport, on the variance of these properties and, specifically, on the layer considered (Ganachaud, 1999; Machín, 2003). The uncertainties for salt and heat anomaly equations follow this equation (Ganachaud, 1999; Machín, 2003): $R_{nn}(Cq) = a*var(C_q)*R_{nn}(mass(q))$ where $R_{nn}(Cq)$ is the uncertainty in the anomaly equation of the property (salt or heat anomaly);

$var(C_q)$ is the variance of this property; $a$ is a weighting factor (4 in the heat anomaly, 1000 in the salt anomaly and $10^6$ in the total salt anomaly) and $q$ is a given equation corresponding to a given layer. Then, these variability estimates are included in the inverse model as the *a priori* uncertainty on the noise of equations in terms of variances of mass, salt anomaly and heat anomaly transports.

On the other hand, the variance of the velocities in the reference level is used as a measure of the *a priori* uncertainty for the

30 unknowns. These variances are calculated from the GLORYS annual mean velocity. However, Machín et al. (2006) concluded that the final mass imbalance is quite independent on both the reference level considered and also on the *a priori* uncertainties in the reference level velocities.

The initial Ekman transports were estimated from the average wind stress during the days of the cruise. A 50% uncertainty was assigned to the initial estimate of Ekman transports, related to the errors in their measurements and to the variability of





the wind stress. An uncertainty of 50 % of the initial value of the freshwater flux, which is 0.0935 Sv, was also prescribed (Ganachaud, 1999; Hernández-Guerra et al., 2005; Machín et al., 2006). Both the Ekman transports, the freshwater flux and their uncertainties were added to the inverse model in the shallowest layer for mass and salt anomaly, and also in the total mass transport and total salt anomaly transport equations.

Dianeutral transfers between the layers were considered to be negligible as compared to other sources of lateral transports, so they were not included in the inverse model. Furthermore, the inverse model provides information only from the box boundaries and cannot be used to provide any details of dianeutral fluxes spatial distribution within the box (Burgoa et al., 2020).

## 3   RESULTS

### 3.1   Hydrography and water masses

$\Theta - S_A$ diagrams exhibit four regions delimited by potential density anomaly contours of 26.42, 27.31 and 27.798 $\mathrm{kg\,m^{-3}}$ (Fig. A3). These isopicnals are equivalent to isoneutrals 26.46, 27.40, and 27.922 $\mathrm{kg\,m^{-3}}$, and separate surface (SW), central (CW), intermediate (IW) and deep water masses (DW) at 119, 698 and 1671 m average depth (Fig. A2). The main water masses sampled during FLUXES-I were NACW and SACW below the mixing layer and above 700 m with $\gamma_n$ values between 26.46 and 27.40 $\mathrm{kg\,m^{-3}}$; modified AAIW and Mediterranean Water (MW) from 700 up to 1700 m with $\gamma_n$ values between 27.40

and 27.922 $\mathrm{kg\,m^{-3}}$; and just the upper part of the North Atlantic Deep Water (NADW) below 1600 m with $\gamma_n$ values higher than 27.922 $\mathrm{kg\,m^{-3}}$ (Zenk et al., 1991; Martínez-Marrero et al., 2008; Pastor et al., 2012).

Straight lines represent the $\Theta - S_A$ relationship between the salty and warm NACW on the northern side of the front and the fresh and cold SACW at its southern side, a distribution similar to that proposed by Tomczak (1981). The main water mass in transects N and W was NACW, while SACW was dominant in transect S. Both NACW and SACW appeared along transect

E. On the other hand, water masses were also well distinguished at intermediate levels, with colder and fresher AAIW over warmer and saltier MW. MW was sampled mainly in transect N and slightly in transects E and W, while AAIW was the main intermediate water sampled in the remaining transect S (Fig. A3).

In vertical sections, the x-axis direction is selected according to the path followed by the vessel during the cruise, starting in the northeast, running along the north, west, south and east sections to end up in the northeast corner again. The N/W, W/S

and S/E corners are indicated with three vertical pink lines at stations 12, 19 and 28, respectively (Figs. A2, A4, A5, A6 and A7). The upper panel in Figure A2 shows the $\gamma_n$ section estimated only with the CTD dataset while the lower panel shows the $\gamma_n$ field until 2000 m once CTD and XBT temperature fields are merged and salinity field is included by interpolating the salinity profiles at XBT positions. This last high resolution $\gamma_n$ field is the one used to calculate the mass transport. The $\gamma_n$ section shows the isoneutrals which define the 8 layers in the model (Fig. A2). The upper four layers grouped surface (above

26.46 $\mathrm{kg\,m^{-3}}$) and central waters. The intermediate waters located between 27.40 and 27.922 $\mathrm{kg\,m^{-3}}$ flowed along the next three layers, while the upper deep waters were accounted in the deepest layer below 27.922 $\mathrm{kg\,m^{-3}}$.

$CT$ and $S_A$ depth/$\gamma_n$ sections are shown in Figure A4. The sections with only CTD data are shown in the upper two panels of Figure A4, while the high resolution depth sections of temperature and salinity fields are shown in the central two panels. The





high resolution sections of $CT$ and $S_A$ with respect to $\gamma_n$ are also shown until the isoneutral of $27.975\ \mathrm{kg\,m^{-3}}$ in the lower two panels. The differences of $CT$ and $S_A$ observed between the water masses at the $\Theta - S_A$ diagrams are verified in the sections of $CT$ and $S_A$, where the water masses can be defined and identified along the sections. At central levels, SACW was mainly sampled in the south-east corner with around $35.65\ \mathrm{g\,kg^{-1}}$ of $S_A$ at $300\ \mathrm{m}$. In contrast, NACW was found in transects N and

W with around $36.15\ \mathrm{g\,kg^{-1}}$ of $S_A$ at $300\ \mathrm{m}$. On the other hand, CVFZ was located where isohaline of $S_A = 36.15\ \mathrm{g\,kg^{-1}}$, equivalent to isohaline of 36 of $S_P$, intersects the $150\ \mathrm{m}$ isobath (Zenk et al., 1991; Burgoa et al., 2020) (Fig. A4). The CVF was crossed in transects S and E during FLUXES-I. First, CVFZ was intersected between stations 23 and 24 where there were important differences in water characteristics, $\Delta\,S_A > 0.70\ \mathrm{g\,kg^{-1}}$ and $\Delta\,CT > 1.92\ ^\circ\mathrm{C}$, between two sides of CVF at $150\ \mathrm{m}$ and then, between stations 33 and 34 with $\Delta\,S_A > 0.30\ \mathrm{g\,kg^{-1}}$ and $\Delta\,CT > 1.10\ ^\circ\mathrm{C}$ at $150\ \mathrm{m}$. At intermediate levels, AAIW

was centered in transect S with a well-marked core below $35.15\ \mathrm{g\,kg^{-1}}$, while the salty MW was sampled at transect N and also slightly in the northern part of transects E and W.

    $O_2$, $NO_3$, $PO_4$ and $SiO_4H_4$ fields interpolated with DIVA are shown in Figures A5 and A6. Between 100 and $800\ \mathrm{m}$ depth, dissolved oxygen has a significant minimum concentration of $60\text{-}90\ \mathrm{\mu mol\,kg^{-1}}$. Between 350 and $1000\ \mathrm{m}$, $NO_3$ and $PO_4$ have maximum concentrations, around $30\text{-}32\ \mathrm{\mu mol\,kg^{-1}}$ and $2\ \mathrm{\mu mol\,kg^{-1}}$, respectively. Finally, $SiO_4H_4$ concentration increases

continuously down to the bottom where it reaches its maximum values with concentrations higher than $25\ \mathrm{\mu mol\,kg^{-1}}$.

    On the other hand, the distributions of $O_2$, $NO_3$, $PO_4$ and $SiO_4H_4$ are related to the location of the different water masses (Figs. A5 and A6). In general, at CW layers of transects N and W where NACW is found, $O_2$ concentrations were higher than in transects S and E where SACW is found, as revealed by $O_2$ concentrations lower than $60\ \mathrm{\mu mol\,kg^{-1}}$ at $300\ \mathrm{m}$ depth. In contrast, the concentrations of the 3 inorganic nutrients in transects S and E were higher than in transects N and W, related

to NACW. At $300\ \mathrm{m}$ depth of transects S and E, concentrations around $27\text{-}30\ \mathrm{\mu mol\,kg^{-1}}$ of $NO_3$, $1.5\text{-}1.7\ \mathrm{\mu mol\,kg^{-1}}$ of $PO_4$ and $7.5\text{-}9.9\ \mathrm{\mu mol\,kg^{-1}}$ of $SiO_4H_4$ were observed. At IW levels the distribution of $O_2$ was quite regular, except in the first layer where low values of $O_2$ in transects S and E were recorded. With respect to inorganic nutrients, transect N had lower concentrations of $NO_3$, $PO_4$ and $SiO_4H_4$ than in the other transects with a higher content of AAIW. Figures A5 and A6 show concentrations higher than 33, 2.05 and $21.4\ \mathrm{\mu mol\,kg^{-1}}$ of $NO_3$, $PO_4$ and $SiO_4H_4$, respectively, associated to AAIW. In the

deepest layer, high concentrations of $O_2$ and inorganic nutrients were found. In the case of $SiO_4H_4$, the concentrations were the highest of the water column while in the $NO_3$ and $PO_4$ the concentrations were lower than the maximum values observed between 400 and $1000\ \mathrm{m}$.

    An intrathermocline anticyclonic eddy was observed in station 4. This eddy is noticeable in the vertical sections of $CT$, $S_A$, $O_2$, $NO_3$ and $PO_4$. On the other side, the intersections of the CVFZ in transects S and E were observed in the vertical sections

of $SiO_4H_4$, $CT$, $S_A$, $O_2$, $NO_3$ and $PO_4$ (Figs. A4, A5 and A6). All these variables change their magnitude as soon as the CVFZ is crossed. In particular, $O_2$ presented two marked minimum values of $60\ \mathrm{\mu mol\,kg^{-1}}$ between 100 and $150\ \mathrm{m}$ when the frontal area was crossed (Fig. A5). These values match their distribution with the local maximum values of $NO_3$, $PO_4$ and $SiO_4H_4$ located just below these $O_2$ minimum values (Figs. A5 and A6).





## 3.2 Absolute velocity

The absolute velocity field perpendicular to each transect and with sign depending on geographic criteria (positive sign is northward and eastward) was estimated with the reference level velocity at $27.962\,\mathrm{kg\,m^{-3}}$ provided by the inversion in transects N, W, and S and, by the annual mean extracted from GLORYS in transect E (Fig. A7). This velocity distribution was validated

comparing the average of the SSH with the mass transport in the shallowest layer of each transect (red bars in Fig. A8). Before validation, the high variability in the area and the time spent in its sampling was taken into account: transect N was carried out from July 14 to 21, transect W from July 21 to 26, transect S from July 26 to August 3 and transect E from August 3 to 8. Hence, the average of SSH during the sampling time period of each transect was selected to analyze the mesoscalar structures (Fig. A8). Besides the intrathermocline anticyclonic eddy centered in station 4, another cyclonic eddy was centered in station 5

next to the first eddy. These two eddies are linked in transect N, the anticyclonic one between stations 3 and 5 and the cyclonic one between stations 4 and 6. Moreover, transect N was crossed by another anticyclonic eddy between stations 10 and 11. On the other hand, the main structures sampled in transect W were a meander between stations 16 and 17 and also a filament found between stations 18 and 19, which actually entered the box through transect S. A second part of this filament, that entered between stations 19 and 21, recirculated and left between stations 20 and 22. From stations 22 to 26, there are some additional

meandering displaced in time and space with respect to the observed structures in the southern area of the SSH corresponding to July 31th. The cyclonic eddy had a significant negative velocity in the last part of the transect S, and it was observed both in the vertical sections of geostrophic velocity and SSH image on July 31th. The velocities in transect E were the smallest and shallowest ones, what makes difficult to identify the structures and link them with altimetry features considering also that the upwelling system and the continental slope are next to the transect (Figs. A7 and A8).

## 3.3 Mass, inorganic nutrients and $O_2$ transports

The mass transports integrated per layer and transect are represented in units of $10^9\,\mathrm{kg\,s^{-1}}$ (equivalent to Sv) in Figure A9. These transports integrated for the different water levels of each transect are also compiled in Table A1. Positive/negative transport values indicate outward/inward transports from/to the closed box. In the shallowest two layers, $-2.5\pm0.6$ and $-0.2\pm0.7$ Sv entered through transects N and E and, $3.8\pm0.8$ and $1.2\pm0.5$ Sv left through transects W and S, respectively. However,

in the next two layers the mass transports reversed their direction in transects N and S. In these two deepest CW layers, the largest mass transport input, $-1.0\pm1.3$ Sv, was from transect E, whereas the largest output continued to be through transect W. In the three layers of IW and in the layer of DW, the direction of mass transport in each transect was rather similar: $-2.4\pm1.6$ Sv came from south and $-1.2\pm2.2$ Sv from east, while $1.8\pm2.1$ and $0.9\pm1.4$ Sv left the box through the transects W and N, respectively (Fig. A9 and Tab. A1).

The transports of $O_2$, $NO_3$, $PO_4$ and $SiO_4H_4$ in $\mathrm{kmol\,s^{-1}}$ are shown in Figure A10 and collected in Table A2. The transport of $O_2$ entered through transects N and E and it left mainly through transect W and to a lesser extent through transect S at SW levels. However, the transports of the three inorganic nutrients at SW levels were small through transects N and E in spite of outward transports through transects W and S had significant values. In the shallowest CW layer, $-200\pm74$, $-15\pm6$,



$-0.95 \pm 0.40$ and $-6 \pm 2\,\mathrm{kmol\,s^{-1}}$ of $O_2$, $NO_3$, $PO_4$ and $SiO_4H_4$ entered from N and left mainly westward with $125 \pm 51$, $34 \pm 14$, $2.00 \pm 0.80$ and $12 \pm 5\,\mathrm{kmol\,s^{-1}}$ of $O_2$, $NO_3$, $PO_4$ and $SiO_4H_4$, respectively. In the second CW layer, transports of $O_2$ entered from transect N and E and left mainly through transect W. The incoming transports of inorganic nutrients in the second CW layer was negligible despite they showed outward transports through transects W, N and S. In the deepest CW

layer where inorganic nutrients accumulated, especially $NO_3$ and $PO_4$, the highest inward inorganic nutrients transport was obtained from S and E with values up to $-38 \pm 33$, $-2.2 \pm 1.9$ and $-33 \pm 29\,\mathrm{kmol\,s^{-1}}$ of $NO_3$, $PO_4$ and $SiO_4H_4$, respectively. Despite the third CW layer is a layer with relative low $O_2$ concentrations, there was also an important inward transport of $-140 \pm 124\,\mathrm{kmol\,s^{-1}}$ of $O_2$ in transects S and E. On the other hand, it is noted a significant outward transport of $31 \pm 42$, $2.20 \pm 2.95$ and $23 \pm 31\,\mathrm{kmol\,s^{-1}}$ of $NO_3$, $PO_4$ and $SiO_4H_4$ northward and westward in the deepest CW layer. In addition, in

the shallowest IW layer where inorganic nutrient concentrations are high and $O_2$ concentration is relatively low, the transports were more intense than in the deepest CW layer: $-220 \pm 175$, $-51 \pm 41$, $-3.2 \pm 2.6$ and $-31 \pm 25\,\mathrm{kmol\,s^{-1}}$ entered from S and E and they left northward and westward with $175 \pm 165$, $40 \pm 38$, $2.75 \pm 2.60$ and $30 \pm 28\,\mathrm{kmol\,s^{-1}}$ of $O_2$, $NO_3$, $PO_4$ and $SiO_4H_4$, respectively. Finally, from the second IW layer to the deepest layer, almost all transport of $O_2$ and inorganic nutrients entered from S and left mainly through the W transect (Fig. A10).

## 15   4   DISCUSSION AND CONCLUSIONS

In this work, the dynamics related to the water masses and their $O_2$ and inorganic nutrient concentrations in the limit between the eastern NASG and the NATG during the summer are analyzed. The sampled water masses in the northern area of the CVB during July-August of 2017 match with those documented so far and their spatial distributions (Hernández-Guerra et al., 2005; Pastor et al., 2012; Peña-Izquierdo et al., 2012; Burgoa et al., 2020). $\Theta - S_A$ diagrams from the cruise FLUXES-I exhibit a

latitudinal change between SACW and NACW, from south to north, below the mixing layer and above 700 m (Pelegrí et al., 2017). At IW levels a second latitudinal change is observed between AAIW and MW from south to north (Zenk et al., 1991), with better defined cores: AAIW cores occupy the upper layers whereas MW cores does the lower layers (Machín and Pelegrí, 2009).

While transects N and S present well defined water masses, transects W and E reflect a transition between transects N and S.

In fact, transect E presents the highest variability in the domain. The variability at SW and CW levels could be related with the position of the CVF, with the proximity of the CUF in the north of the domain and with the meso- and sub-mesoscale structures associated to these two frontal systems, as the upwelling filaments off Cape Blanc (Meunier et al., 2012; Lovecchio et al., 2018). At IW levels, some variability was observed in transects N, W and E due to the MW. However, AAIW is well defined with properties rather constant throughout transect S.

The characteristic of the water masses are conditioned by their origin and by the path followed in their way to the CVB. Their properties are summarized in Figures A11 and A12 where the relationships between *in situ* measurements of $S_A$, $O_2$, $NO_3$ and $PO_4$ are displayed. SACW is sampled in transect S and is characterized by relatively low $O_2$ and high inorganic nutrients concentrations due to its old age (third column of Fig. A11). On the other hand, NACW is a saltier water mass with low

concentrations of $NO_3$ and $PO_4$ and high of dissolved $O_2$. At least in this summer 2017, SACW was not confined above $\gamma_n = 26.85 - 26.86$ kg m$^{-3}$ (equivalent to $\sigma_\theta = 27.8$ kg m$^{-3}$), in the upper CW levels, and NACW was located below it in the lower CW levels, as previously reported (Voituriez and Chuchla, 1978; Peña-Izquierdo et al., 2015). In transects W and E, especially in E, properties of both NACW and SACW were clearly distinct and coexist below $\gamma_n = 26.85$ kg m$^{-3}$ (350 m depth). This

fact is observed in Figure A11, especially in charts of dissolved oxygen versus $S_A$, dissolved oxygen versus $NO_3$ and dissolved oxygen versus $PO_4$ at transect E, where two "patches" are shown, one on top of the other, corresponding to NACW (up) and SACW (down). Additionally, the NACW and SACW property distributions shown in Figure A11 compare well with those shown by Pastor et al. (2008) and Pelegrí and Benazzouz (2015). The shallowest SACW (between 26.46 and 26.85 kg m$^{-3}$) in transect S had concentrations of $NO_3$ and $PO_4$ higher than 12.5 and 0.9 µmol kg$^{-1}$ and lower than 36.5 g kg$^{-1}$ and

115 µmol kg$^{-1}$ of $S_A$ and $O_2$. In contrast, the shallowest NACW (in the same range of $\gamma_n$ of shallowest SACW) in transect N had concentrations of $NO_3$ and $PO_4$ lower than 12.5 and 0.9 µmol kg$^{-1}$ and higher than 36.5 g kg$^{-1}$ and 115 µmol kg$^{-1}$ of $S_A$ and $O_2$. In the same way, the deepest SACW/NACW in transect S/N had concentrations of $NO_3$ and $PO_4$ higher/lower than 30 and 2 µmol kg$^{-1}$ and lower/higher than 35.25 g kg$^{-1}$ and 95 µmol kg$^{-1}$ of $S_A$ and $O_2$ (Fig. A11).

At IW, between 27.40 and 27.922 kg m$^{-3}$, transect S was also the one with the least variability due to the predominance

of AAIW. In contrast, transects E and especially N and W had a larger variability due to the coexistence of AAIW and MW, represented by sudden salinity increases especially in charts of dissolved oxygen versus $S_A$ up to more than 27.7 kg m$^{-3}$ (Fig. A12). AAIW had the properties well defined in transect S: the shallowest AAIW had $S_A$ lower than 35.18 g kg$^{-1}$ and concentrations of $NO_3$, $PO_4$ and $O_2$ between 32-37, 2.1-2.3 and 70-105 µmol kg$^{-1}$, while the deepest AAIW had $S_A$ between 35.15-35.23 g kg$^{-1}$ and concentrations of $NO_3$, $PO_4$ and $O_2$ between 22-26, 1.5-1.7 and 190-215 µmol kg$^{-1}$, respectively.

However, MW presents some variability in its properties, in accordance with the ranges documented by Burgoa et al. (2020).

The shadow zone proposed by Luyten et al. (1983) between well-ventilated NASG and less-ventilated NATG next to a highly productive coastal upwelling system leads to the development of an oxygen minimum zone (OMZ) within CW and IW levels (Karstensen et al., 2008). In our domain, the OMZ is centered in transect S and it is located between 100 and 800 m depth with its core around 400 m between isoneutrals 27.1 and 27.3 kg m$^{-3}$. That distribution matches well with the one provided

by Peña-Izquierdo et al. (2015) and it has high concentrations of $NO_3$ and $PO_4$. In addition, below the location of the CVF in transects S and E, the sampled minimum values of $O_2$ and maximum values of $NO_3$ and $PO_4$ indicate a local remineralization just under the front at 100-150 m.

As soon as the CVF is crossed, especially in transect E, a sudden and abrupt change due to the coexistence of NACW and SACW was observed. The predominance/lack of SACW/NACW south of the front in transect S suggests that CVF functions

as a barrier against lateral transports across the front. On the other hand, the analyzes performed here cannot assess a lateral transport along the CVF.

In summer 2017, in the second CW layer between 26.85 and 27.162 kg m$^{-3}$, an inversion of transport was observed in transects N, S and E. This inversion at about 300 m depth in transects N and E and approximately 450 m depth in transect S had been observed in temperature and salinity before (Voituriez and Chuchla, 1978; Pastor et al., 2012; Peña-Izquierdo et al., 2012;

Peña-Izquierdo et al., 2015). Some of these authors relate it to different circulation patterns above and below the inversion that





we have also tried to analyze. In the first two layers of the water column, which group SW and the shallowest CW, incoming transports of $-2.5 \pm 0.6$ and $-0.2 \pm 0.7$ Sv were estimated in transects N and E while outward transports of $3.8 \pm 0.8$ and $1.2 \pm 0.5$ Sv were obtained in transects W and S. That is, in the first two layers, approximately 90% of the transport that entered the domain and came from the north, left the domain in transect W with 75% directed to the open ocean. In addition,

north of Cape Blanc, this transport can be driven, at least in the first layer, by the wind field (Benazzouz et al., 2014). On the other hand, in the two deepest CW layers the circulation patterns change with a significant input of $-1.0 \pm 1.3$ Sv from the east. In these two layers, where the inversion is given between 26.85 and 27.162 $\mathrm{kg\,m^{-3}}$, the main circulation was also westward with a slight northward transport increasing in transect N. These estimated circulation patterns in CW coincide with the climatological streamlines of the geostrophic field estimated from WOA18 for the summer season (Fig. A13). In transects

N and W, a west-southwest circulation of NACW was observed throughout the entire water column. However, the cyclonic northward circulation that intensifies in the deepest CW layer produced that only SACW was found in transect S and it also induced the coexistence between NACW and SACW in transect E (Fig. A13). On the other hand, at IW and DW layers, the main transport comes from the south with $-2.4 \pm 1.6$ Sv. This northward transport starts to be noticed in the last CW layer, specially in transects E and S.

In spite of the 21.31% of the relative error associated to the imbalance of $2.6 \pm 4.5$ Sv (Tab. A1) mainly related with the lack of sypnopticity in the sampling, more than 60% of the mass transport occurred in SW and CW layers, with the exception of transect S where transport barely exceeds 45%. Transports perpendicular to the African coast were 10% more intense than north-south/south-north transports at SW and CW levels. In general, around 70-75% of the mass transport were westward perpendicular to the African coast at SW and CW levels (Fig. A14). More than 75% of the southward mass transport in

transect N reduced to a 50% when it crosses transect S at SW and CW levels. In the opposite way, south-north, more than 50% of the mass transport at IW and DW levels moves north in transect S, and it flows northward through transect N with half of its intensity. On the other hand, 20-30% of the mass transport perpendicular to the African coast at IW and DW levels moved away from the coast (Fig. A14).

Despite the CVFZ is in the middle of the study area preventing transports at CW in the north-south direction, the estimated

total transport of $7.4 \pm 2.5$ Sv towards the open ocean is comparable to the $7.0 \pm 2.6$ Sv estimated by Burgoa et al. (2020) for fall 2002 at 26°W just north of this domain. Thus, lateral transport in the domain seems to be only moderately affected by the location of the CVF, in the southeastern corner of the domain with an angle of 45° with respect to the African coast (Fig. A14).

On the other hand, $O_2$ and inorganic nutrients transports are conditioned by their concentrations and the dynamics in each layer. The inorganic nutrient transports had an inversion between 26.85 and 27.162 $\mathrm{kg\,m^{-3}}$ on the second CW layer. This

inversion was less marked and at a greater depth in $O_2$ transport. In the first CW layer, $O_2$ and inorganic nutrients came from the north, but below the inversion they came mainly from the east and south. However, the transport was mainly westward, to the open oligotrophic ocean, in all CW layers. In the third CW layer, there was also important northward transport of inorganic nutrients through transect N. In IW and DW, these transports came from the south and east and they left also mainly westward as with mass transports. The most intense and important transports of $O_2$ and inorganic nutrients occur in the deepest CW

layer and the shallowest two IW layers (Fernández-Castro et al., 2018; Burgoa et al., 2020). In these three layers, $O_2$ enters





with $-280\pm262$ and $-340\pm206\,\mathrm{kmol\,s^{-1}}$ from east and south and it leaves with $95\pm124$ and $255\pm238\,\mathrm{kmol\,s^{-1}}$ northward and westward. In the same way, inorganic nutrients entered with $-58\pm54\,/-3.5\pm3.3\,/-34\pm32$ and $-65\pm39\,/-4.2\pm2.5$ $/-42\pm26\,\mathrm{kmol\,s^{-1}}$ of $NO_3$ / $PO_4$ / $SiO_4H_4$ from east and south and left with $35\pm46$ / $2.40\pm3.13$ / $28\pm37$ and $57\pm53$ / $3.9\pm3.6$ / $40\pm37\,\mathrm{kmol\,s^{-1}}$ of $NO_3$ / $PO_4$/ $SiO_4H_4$ northward and westward. These transports do not match with the lateral

advection ranges of Burgoa et al. (2020) north of our domain where inorganic nutrients transports are smaller than in the CVB but they are in agreement with the climatological ranges reported by Fernández-Castro et al. (2018).

In summary, the differences between NASG and NATG regions which are well defined by $O_2$ and inorganic nutrients concentrations in CW layers and in the shallowest two IW layers, condition the transport of these properties. In the shallowest CW layer, transports of $O_2$ and inorganic nutrients come from the north and they leave mainly westward by the presence of

the CVFZ. The estimated net outflows through the east and south in this layer can be justified by the variability sampled in transects S and E related with the southeastern cyclonic gyre and with meander-like structures along the African slope. In the deepest CW layer and the rest of IW layers, the main inputs of inorganic nutrients from the east and south leave latter the box through the north and west transects with smaller transport values but which can change the properties of the more oligotrophic waters north of CVFZ. These inorganic nutrients transport values in the deepest CW layer and in the rest of IW layers might

be explained, on the one hand, by the local remineralization in the southeastern of the domain, where the OMZ is located; moreover, the transport intensities are higher in transects S and E than in transects N and W. On the other hand, despite the fact that the southeastern waters have lower concentrations of $O_2$ than those of the northwest in the deepest CW layer and the shallowest IW layer, the incoming transports from the east and south are higher than the transports northward and westward. This fact may also be due to a greater intensity of incoming transports from south and east in these two levels and below,

hindering their ventilation during this season.





# Appendix A

## A1

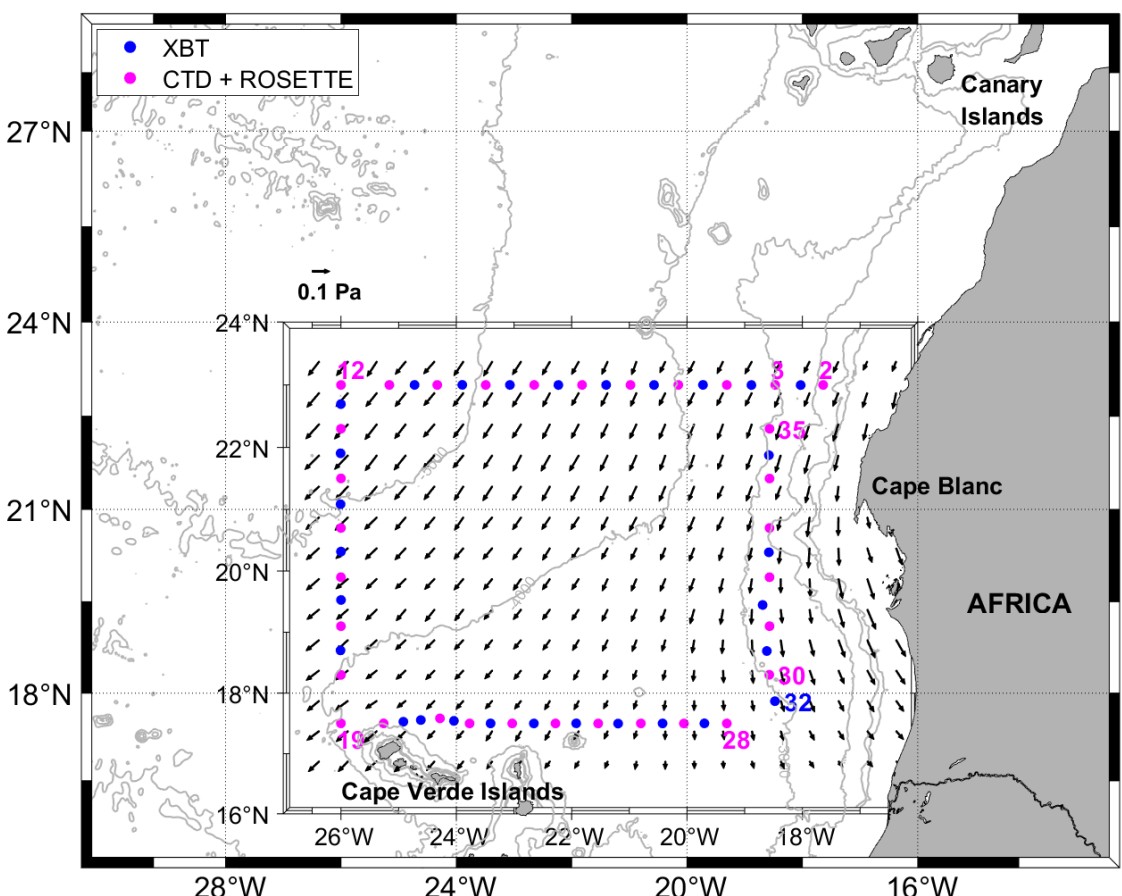

**Figure A1.** CTD-rosette sampling stations (pink dots) and XBT (blue dots) during FLUXES-I cruise. Time-averaged wind stress during the cruise is also represented with the inset arrow denoting the scale (shown with half of the original spatial resolution).





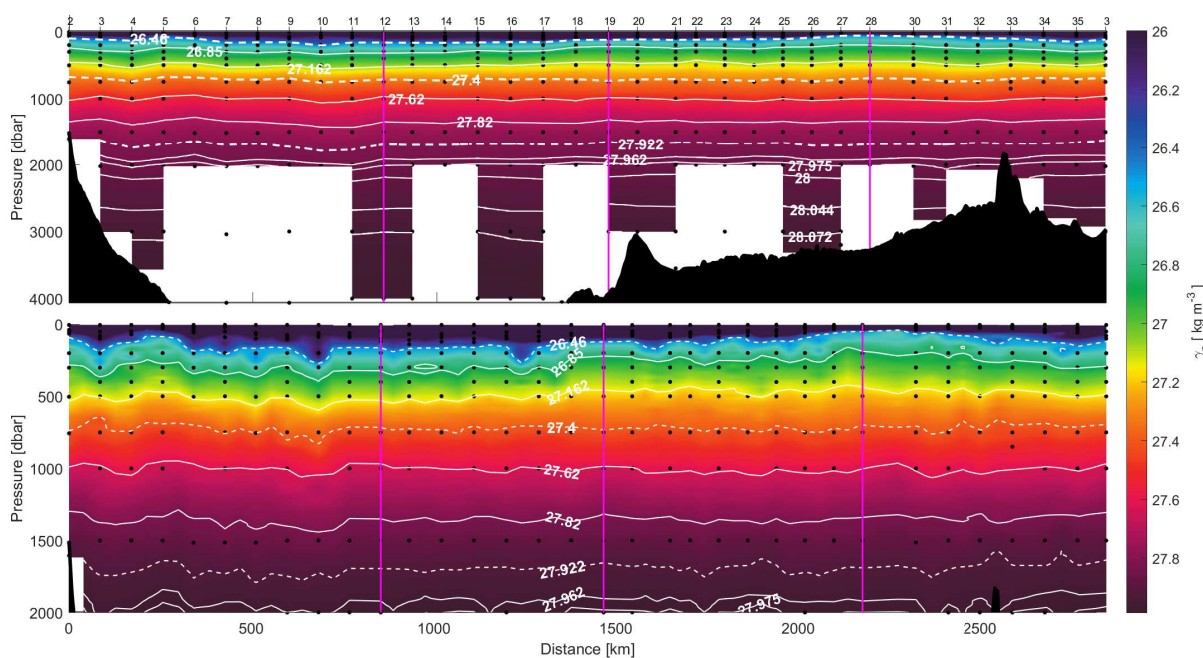

**Figure A2.** $\gamma_n$ vertical sections during FLUXES-I cruise. White dashed isoneutrals limit the different water type layers. The direction chosen for the representation of the transects is the course of the vessel. Distance is calculated with respect to the first station (2). The section is divided into four transects: N transect from east to west (from station 2 to 12), W transect from north to south (from station 12 to 19), S transect from west to east (from stations 19 to 28) and E transect from south to north (from 28 to 3). The 4 transects are separated by three vertical pink lines located at stations number 12, 19 and 28. The sampling points of dissolved oxygen and IN used in this work are represented in black dots. Upper figure shows $\gamma_n$ section estimated only with CTD dataset and lower figure shows the high resolution $\gamma_n$ field estimated until 2000 m from CTD-XBT.

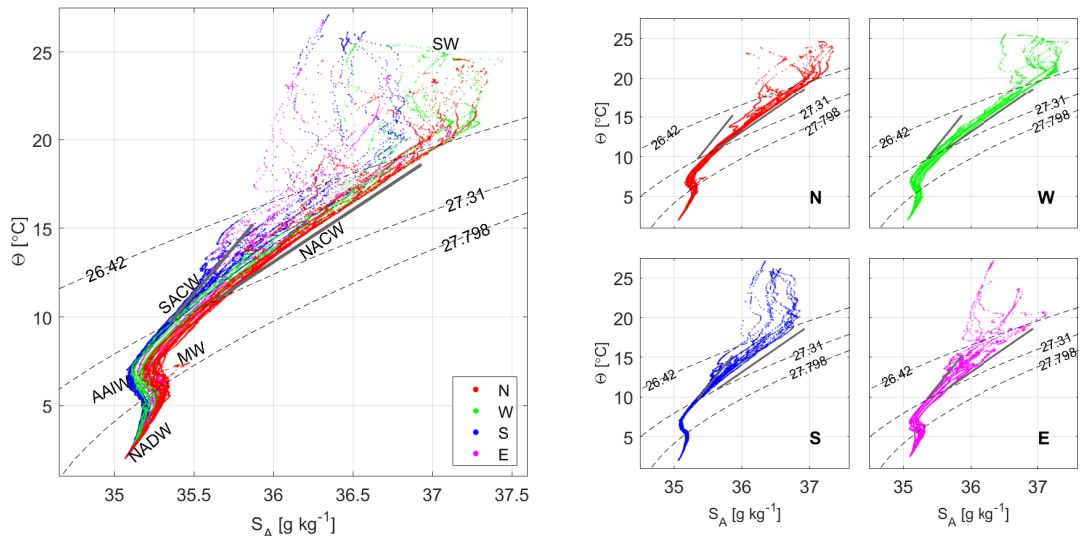

**Figure A3.** $\Theta - S_A$ diagrams during FLUXES-I cruise. The different water masses at northern (N, red dots), western (W, green dots), southern (S, blue dots) and eastern (E, pink dots) transects are surface waters (SW), North Atlantic Central Water (NACW), South Atlantic Central Water (SACW), modified Antarctic Intermediate Water (AAIW), Mediterranean Water (MW) and North Atlantic Deep Water (NADW). Potential density anomaly contours (gray dashed lines) equivalent to 26.46, 27.4 and 27.922 $\mathrm{kg\,m^{-3}}$ isoneutrals delimit the surface, central, intermediate and deep water levels. Straight lines represent the $\Theta - S_A$ relationship for NACW and SACW equivalent to that proposed by Tomczak (1981).



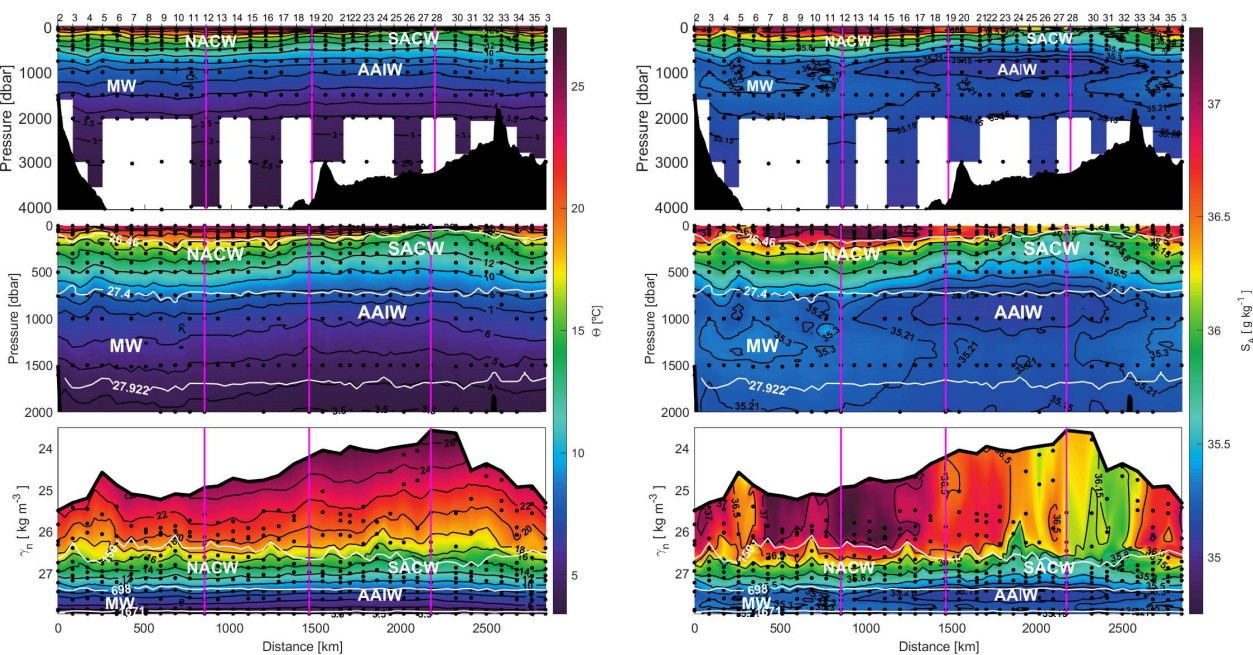

**Figure A4.** Sections of $CT$ (left) and $S_A$ (right) with respect to depth (first and second line) and $\gamma_n$ (third line) during FLUXES-I cruise. The direction chosen for the representation is the same as in Fig. A2. The 4 transects are separated by three vertical pink lines located at stations number 12, 19 and 28. In depth sections, the isoneutrals which delimit the surface, central, intermediate and deep water are represented by white contours. In $\gamma_n$ sections, the depths of 150, 698 and 1671 m are also shown. The sampling points of dissolved oxygen and IN used in this work are represented in black dots. First line sections are estimated only with CTD dataset and second line sections are high resolution estimated until 2000 m from CTD-XBT.



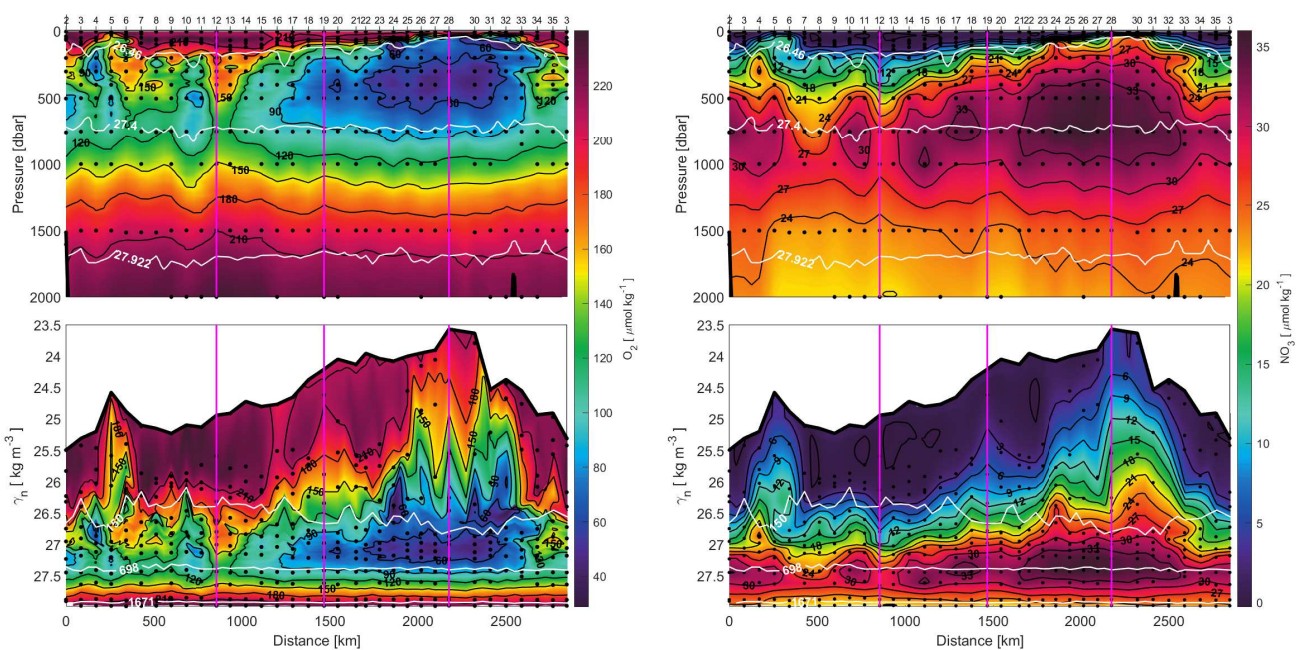

**Figure A5.** Sections of $O_2$ (left) and $NO_3$ (right) with respect to depth (upper line) and $\gamma_n$ (lower line) during FLUXES-I cruise. The direction chosen for the representation is the same as in Fig. A2. The 4 transects are separated by three vertical pink lines located at stations number 12, 19 and 28. In depth sections, the isoneutrals which delimit the surface, central, intermediate and deep water are represented by white contours. In $\gamma_n$ sections, the depths of 150, 698 and 1671 m are also shown. The sampling points of $O_2$ and $NO_3$ used in this work are represented in black dots.



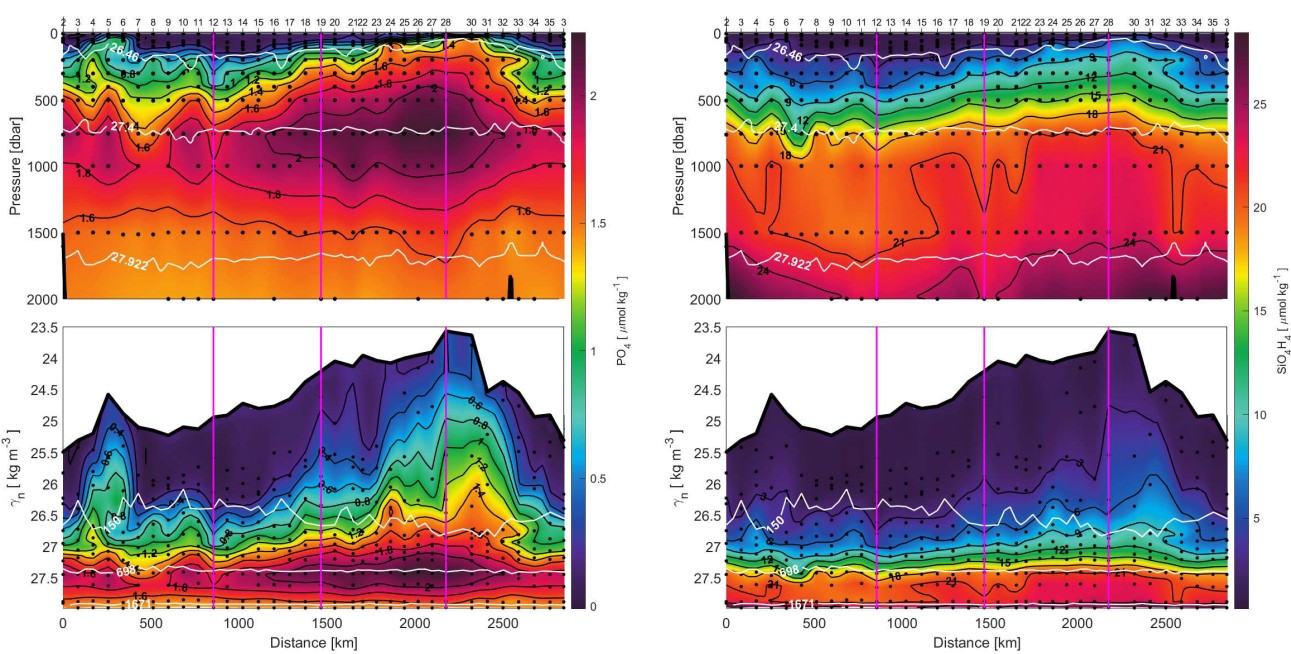

**Figure A6.** Sections of $PO_4$ (left) and $SiO_4H_4$ (right) with respect to depth (upper line) and $\gamma_n$ (lower line) during FLUXES-I cruise. The direction chosen for the representation is the same as in Fig. A2. The 4 transects are separated by three vertical pink lines located at stations number 12, 19 and 28. In depth sections, the isoneutrals which delimit the surface, central, intermediate and deep water are represented by white contours. In $\gamma_n$ sections, the depths of 150, 698 and 1671 m are also shown. The sampling points of $PO_4$ and $SiO_4H_4$ used in this work are represented in black dots.



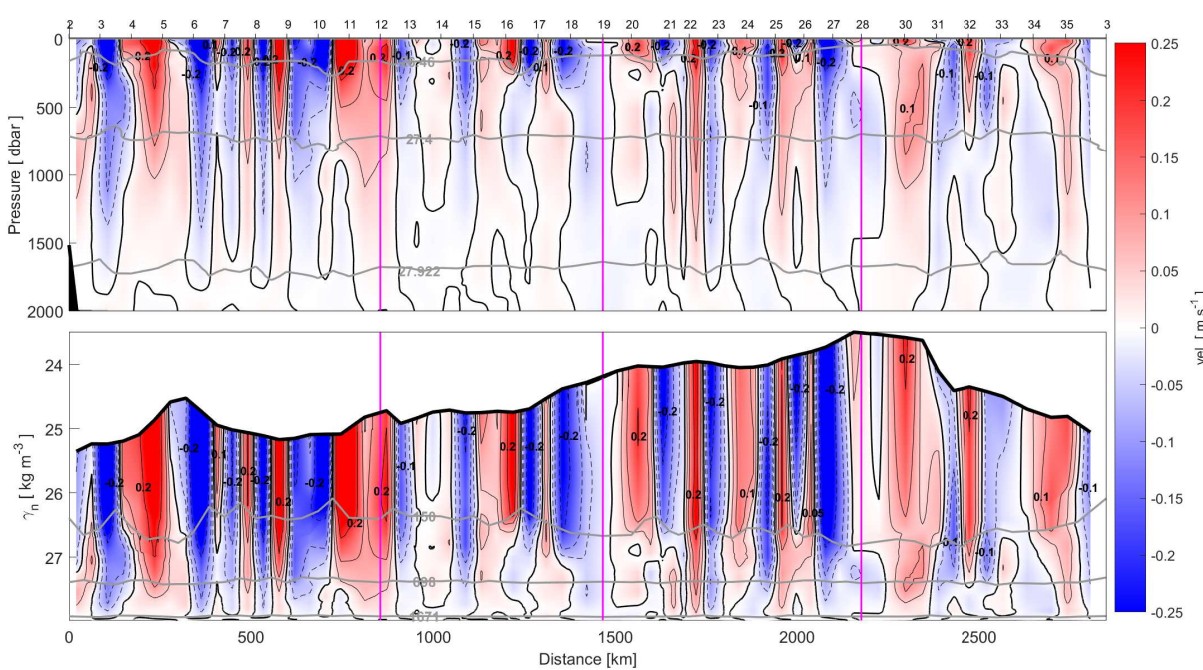

**Figure A7.** Sections of the absolute geostrophic velocity perpendicular to each transect with respect to depth (upper panel) and $\gamma_n$ (lower panel) during FLUXES-I cruise. The velocity sign was selected on geographic criteria (positive sign northward and eastward). The direction chosen for the representation is the same as in Fig. A2. Zero contour line is the widthest black line. The 4 transects are separated by three vertical pink lines located at stations number 12, 19 and 28. In depth sections, the isoneutrals which delimit the surface, central, intermediate and deep water are represented by grey contours. In $\gamma_n$ sections, the depths of 150, 698 and 1671 m are also shown.



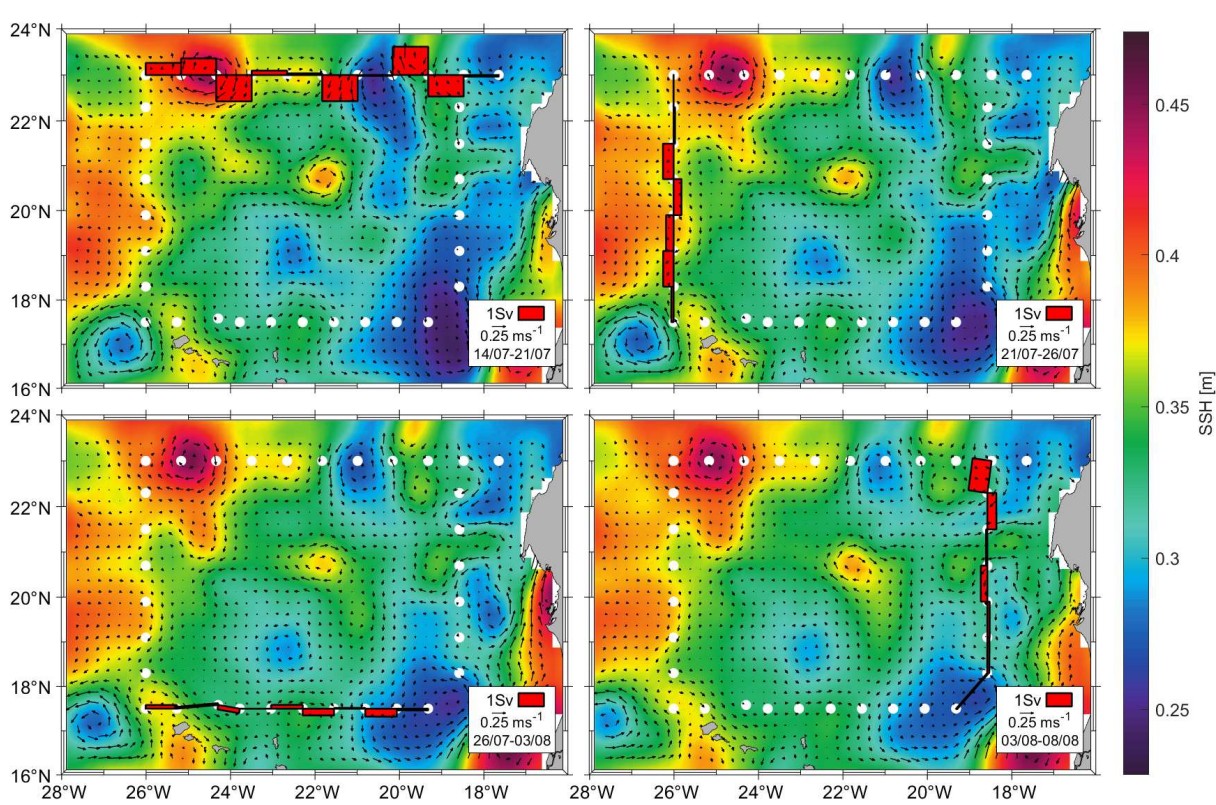

**Figure A8.** Averages of SSH and derived geostrophic velocity from SEALEVEL_GLO_PHY_L4_REP_OBSERVATIONS_008_047 for the sampling time period of each transect: from July 14 to 21 (upper left), from July 21 to 26 (upper right), from July 26 to August 3 (lower left) and from August 3 to 8 (lower right) of 2017. The red bars represent the mass transports in the shallowest layer. White dots are CTD stations of FLUXES-I cruise.



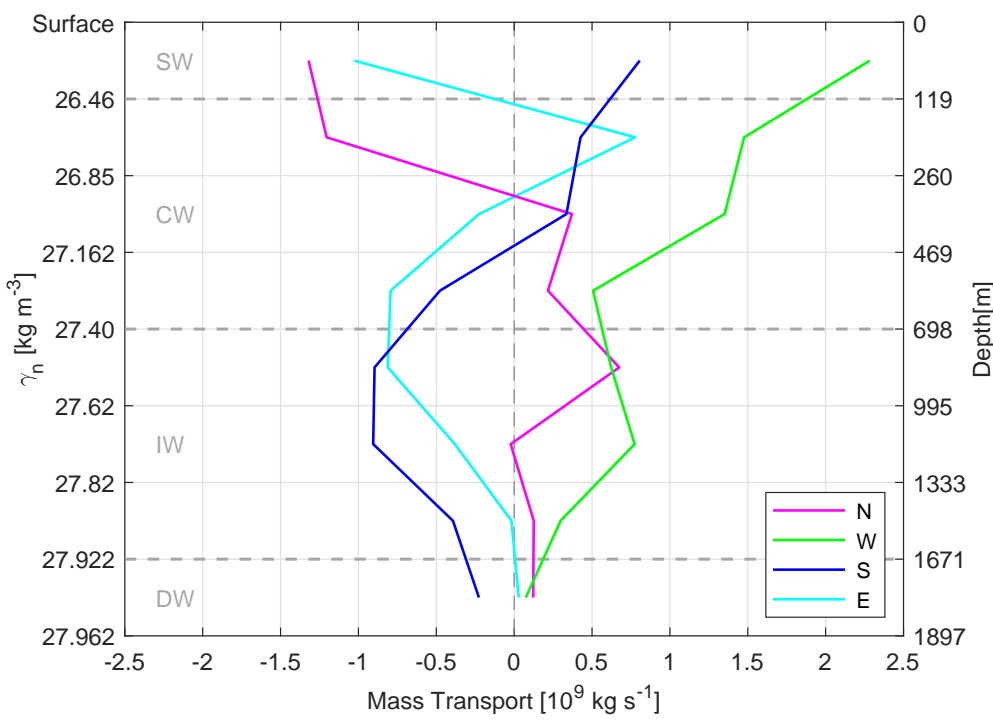

**Figure A9.** Mass transports integrated per transect at north (N, pink line), west (W, green line), south (S, blue line) and east (E, cyan line) transects during FLUXES-I cruise. See Tab. A1 to check mass transport values at each layer per transect. Negative/positive values indicate inward/outward transports.



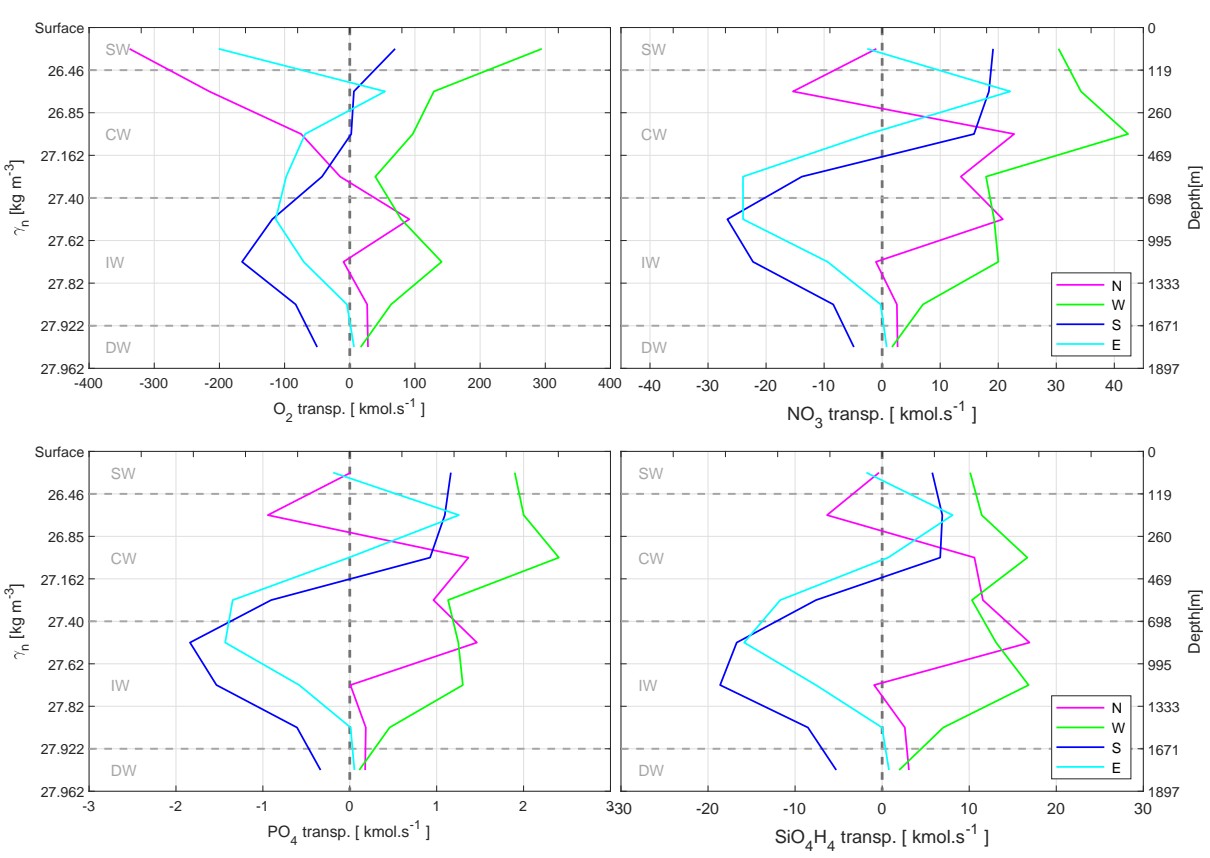

**Figure A10.** $O_2$, $NO_3$, $PO_4$ and $SiO_4H_4$ transports ($\mathrm{kmol\,s^{-1}}$) integrated at transects north (N, pink line), west (W, green line), south (S, blue line) and east (E, cyan line) during FLUXES-I cruise. Negative/positive values indicate inward/outward transports as in Fig. A9.



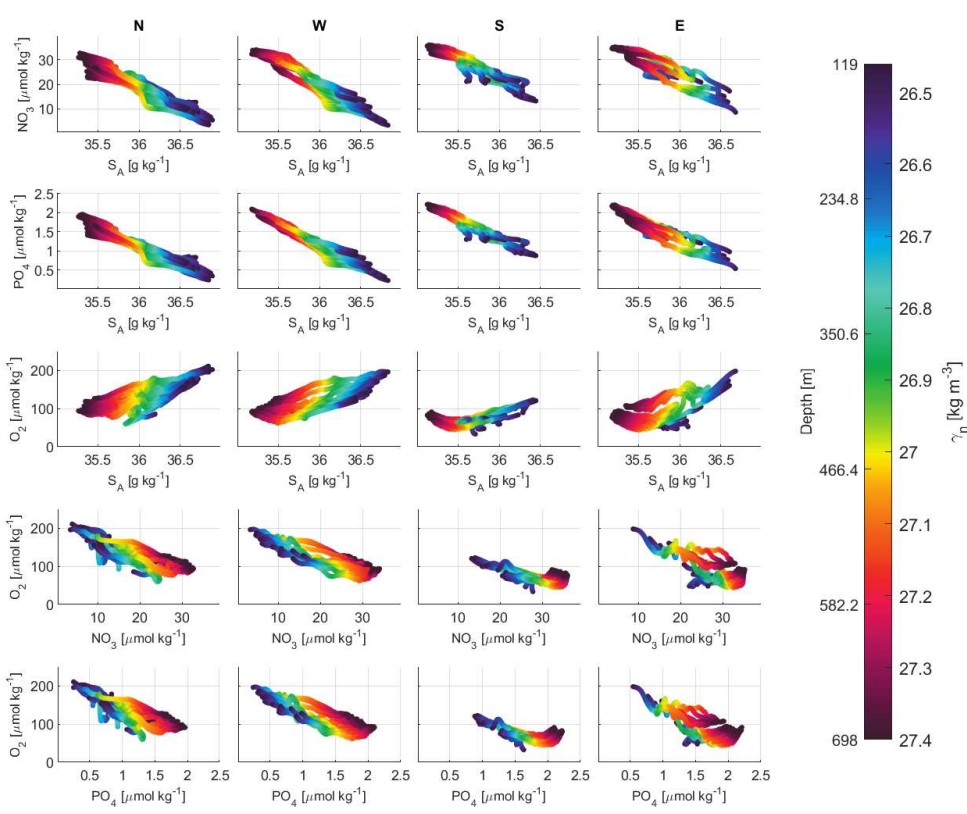

**Figure A11.** Scatter plots of *in situ* observations of $NO_3$ and $PO_4$ and dissolved oxygen ($\mu mol\,kg^{-1}$) with respect to $S_A$ and $\gamma_n$ in transects north (N, first column), west (W, second column), south (S, third column) and east (E, fourth column) for CW layers in first, second and third line. In fourth and fifth lines, scatter plots of $NO_3$ - dissolved oxygen and $NO_3$ - dissolved oxygen in 4 transects and also for CW layers.

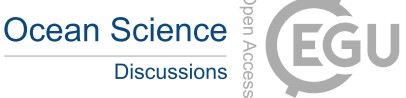



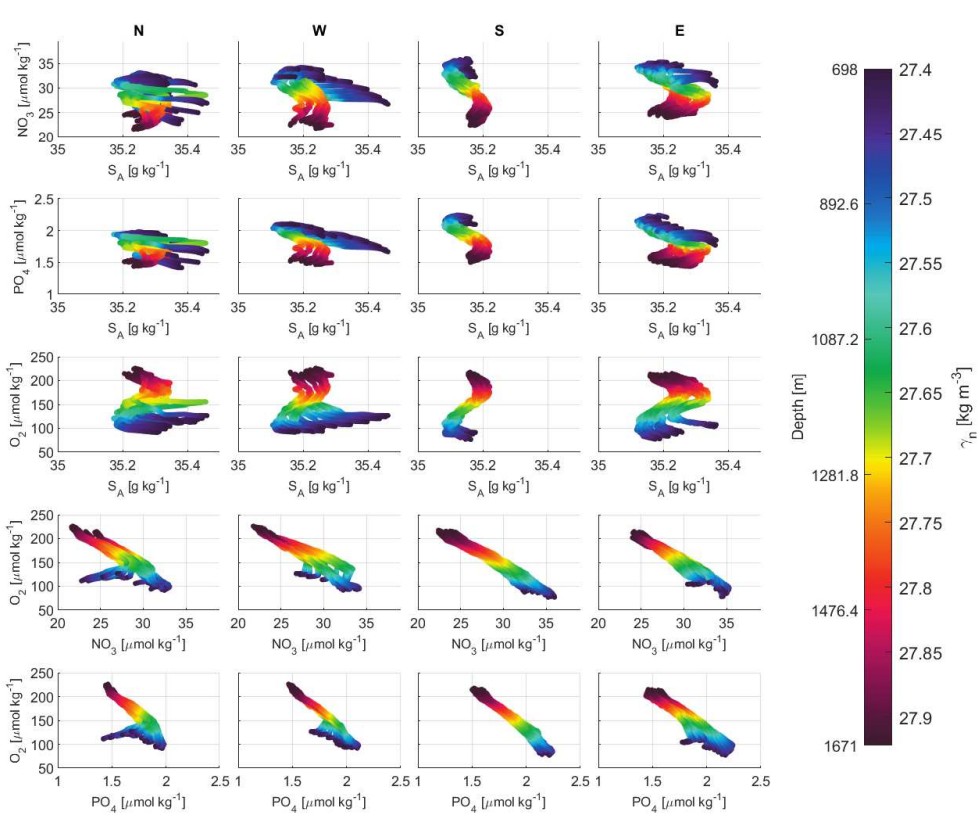

**Figure A12.** Scatter plots of *in situ* observations of $NO_3$ and $PO_4$ and dissolved oxygen in units of $\mu mol\,kg^{-1}$ with respect to $S_A$ and $\gamma_n$ in transects north (N, first column), west (W, second column), south (S, third column) and east (E, fourth column) for IW layers in first, second and third line. In fourth and fifth lines, scatter plots of $NO_3$ - dissolved oxygen and $NO_3$ - dissolved oxygen in 4 transects and also for IW layers.





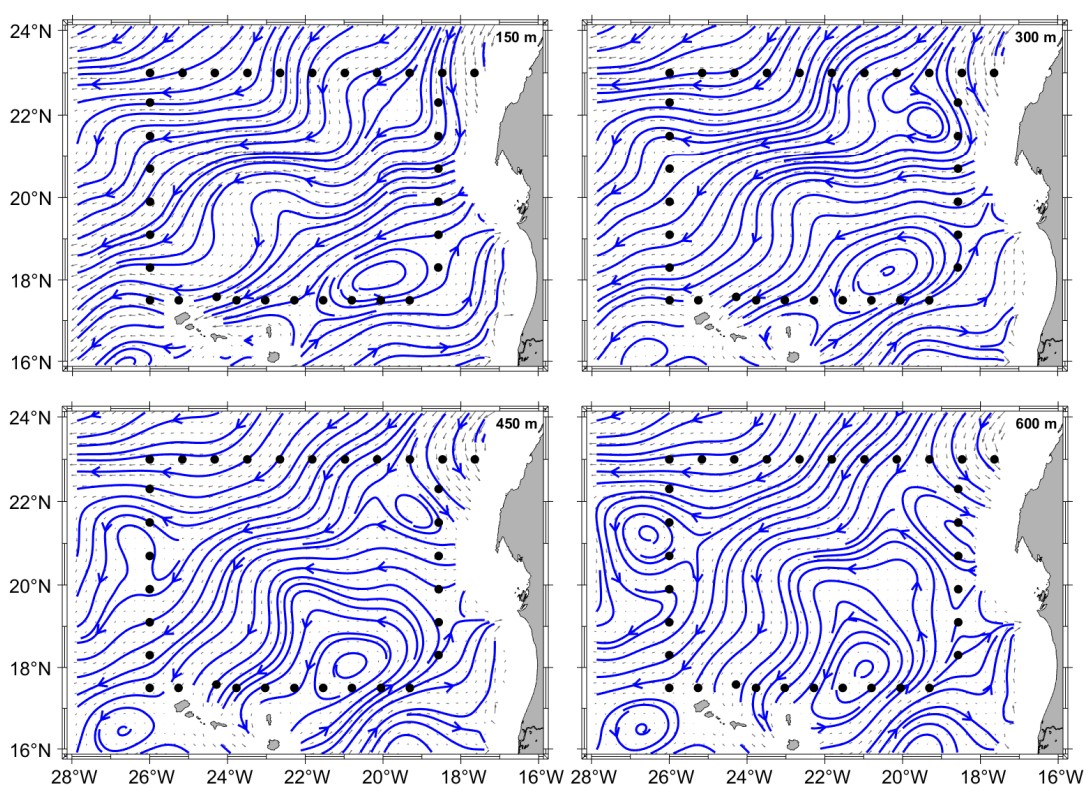

**Figure A13.** Summer climatological geostrophic velocity field estimated from WOA18 with their streamlines. From top to bottom and from left to right: 150, 300, 450 and 600 m. Black dots are CTD stations during FLUXES-I cruise.





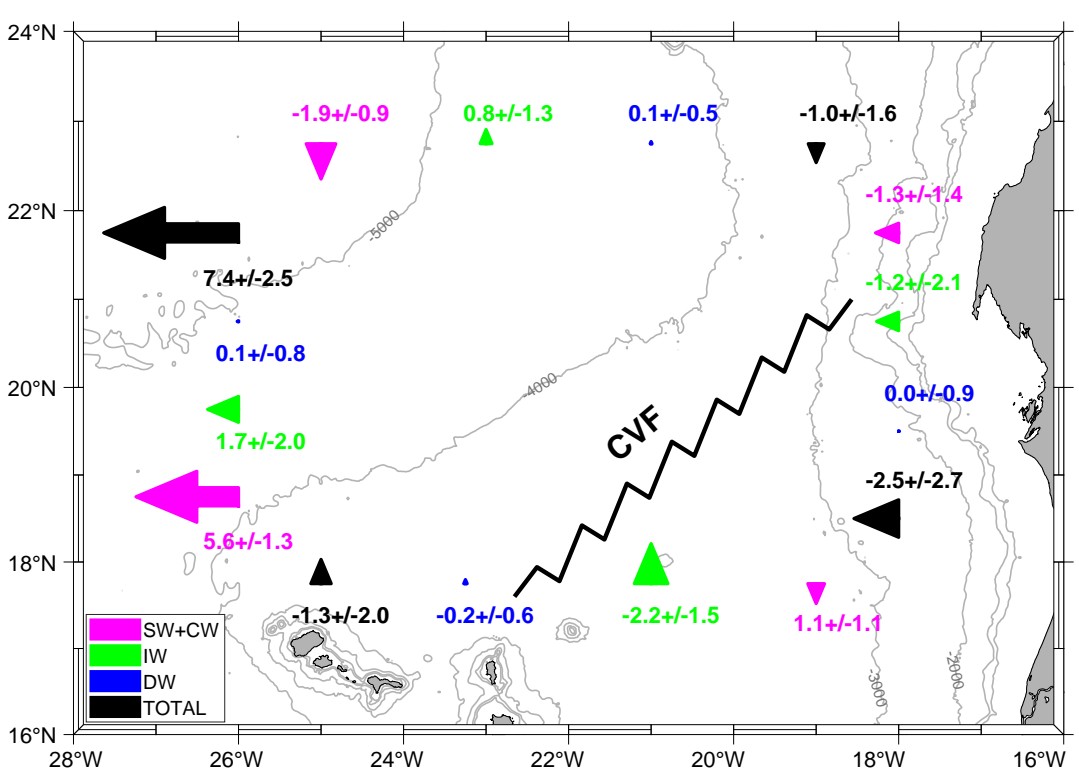

**Figure A14.** Mass transports with their errors (Sv) at surface and central waters (SW+CW, pink arrow), intermediate waters (IW, green arrow) and deep waters (DW, blue arrow) during FLUXES-I cruise. Vertically integrated transport at each transect (TOTAL) is represented with black arrows. Negative/positive values indicate inward/outward transports as in Fig. A9. All arrows are located in positions which optimize their visibility, representing the integrated transports along transect. The pink arrows represent the integrated transports for SW plus CW layers.





**Table A1.** Mass transports with their errors (Sv) for SW, CW, IW, and DW across northern, western, southern and eastern transects for FLUXES-I cruise. Positive/negative values indicate outward/inward transports. The last row is the integrated transport for all the water column. The last column is the disajust or imbalance of each layer.

| Water level | $\gamma_n$ **range** $[\mathrm{kg\,m^{-3}}]$ **Depth range** [m] | North | West | South | East | Imbalance |
|---|---|---|---|---|---|---|
| SW | Surface$-26.46$ Surface$-119$ | $-1.3\pm0.4$ | $2.3\pm0.6$ | $0.8\pm0.4$ | $-1.0\pm0.4$ | $0.7\pm0.9$ |
| CW | $26.46-26.85$ $119-260$ | $-1.2\pm0.4$ | $1.5\pm0.5$ | $0.4\pm0.3$ | $0.8\pm0.6$ | $1.5\pm0.9$ |
| | $26.85-27.162$ $260-469$ | $0.4\pm0.5$ | $1.4\pm0.7$ | $0.3\pm0.6$ | $-0.2\pm0.9$ | $1.8\pm1.4$ |
| | $27.162-27.40$ $469-698$ | $0.2\pm0.5$ | $0.5\pm0.8$ | $-0.5\pm0.7$ | $-0.8\pm0.9$ | $-0.5\pm1.5$ |
| IW | $27.40-27.62$ $698-995$ | $0.7\pm0.7$ | $0.6\pm1.0$ | $-0.9\pm0.8$ | $-0.8\pm1.1$ | $-0.4\pm1.8$ |
| | $27.62-27.82$ $995-1333$ | $-0.0\pm0.8$ | $0.8\pm1.2$ | $-0.9\pm0.9$ | $-0.4\pm1.2$ | $-0.5\pm2.2$ |
| | $27.82-27.922$ $1333-1671$ | $0.1\pm0.8$ | $0.3\pm1.2$ | $-0.4\pm0.9$ | $-0.0\pm1.2$ | $0.0\pm2.0$ |
| DW | $27.922-27.962$ $1671-1897$ | $0.1\pm0.5$ | $0.1\pm0.8$ | $-0.2\pm0.6$ | $0.0\pm0.9$ | $-0.0\pm1.5$ |
| TOTAL | Surface$-27.962$ Surface$-1897$ | $-1.0\pm1.6$ | $7.4\pm2.5$ | $-1.3\pm2.0$ | $-2.5\pm2.7$ | $2.6\pm4.5$ |

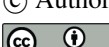



**Table A2.** Transports of $O_2$, $NO_3$, $PO_4$ and $SiO_4H_4$ in $\mathrm{kmol\,s^{-1}}$ with their errors relative to mass transport for SW, CW, IW, and DW across northern, western, southern and eastern transects for FLUXES-I cruise. Positive/negative values indicate outward/inward transports. Total row of each inorganic nutrient or $O_2$ is its integrated transport in all the water column. The last column is the disajust or imbalance of each water level.

| Variable | Water level | North | West | South | East | Imbalance |
|---|---|---|---|---|---|---|
| $O_2$ | SW | $-337.90 \pm 99.40$ | $294.70 \pm 72.50$ | $69.15 \pm 30.10$ | $-201.05 \pm 73.90$ | $-174.70 \pm 146.60$ |
| | CW | $-304.90 \pm 122.20$ | $264.90 \pm 55.85$ | $-34.05 \pm 32.05$ | $-112.00 \pm 118.80$ | $-186.00 \pm 182.20$ |
| | IW | $108.40 \pm 181.40$ | $283.10 \pm 327.85$ | $-367.10 \pm 257.50$ | $-188.75 \pm 322.00$ | $-164.30 \pm 557.10$ |
| | DW | $27.90 \pm 114.50$ | $16.80 \pm 175.90$ | $-50.00 \pm 141.55$ | $6.50 \pm 207.20$ | $1.20 \pm 327.10$ |
| | TOTAL | $-506.40 \pm 802.20$ | $859.50 \pm 289.20$ | $-381.70 \pm 570.80$ | $-495.20 \pm 537.80$ | $-523.85 \pm 1159.00$ |
| $NO_3$ | SW | $-1.00 \pm 0.30$ | $30.40 \pm 7.50$ | $19.10 \pm 8.30$ | $-2.60 \pm 0.90$ | $45.90 \pm 11.20$ |
| | CW | $20.90 \pm 8.40$ | $94.50 \pm 19.90$ | $20.40 \pm 19.20$ | $-4.25 \pm 4.50$ | $131.60 \pm 29.30$ |
| | IW | $22.30 \pm 37.30$ | $46.35 \pm 53.70$ | $-57.30 \pm 40.20$ | $-33.60 \pm 57.35$ | $-22.30 \pm 95.80$ |
| | DW | $2.70 \pm 10.90$ | $1.70 \pm 17.70$ | $-4.90 \pm 13.80$ | $0.80 \pm 24.80$ | $0.25 \pm 35.20$ |
| | TOTAL | $44.90 \pm 71.10$ | $173.00 \pm 58.20$ | $-22.70 \pm 34.00$ | $-39.70 \pm 43.10$ | $155.50 \pm 107.00$ |
| $PO_4$ | SW | $0.00 \pm 0.00$ | $1.90 \pm 0.50$ | $1.20 \pm 0.50$ | $-0.20 \pm 0.10$ | $2.90 \pm 0.70$ |
| | CW | $1.40 \pm 0.60$ | $5.50 \pm 1.20$ | $1.10 \pm 1.10$ | $-0.10 \pm 0.10$ | $7.90 \pm 1.70$ |
| | IW | $1.65 \pm 2.80$ | $3.00 \pm 3.50$ | $-4.00 \pm 2.80$ | $-2.00 \pm 3.40$ | $-1.30 \pm 6.30$ |
| | DW | $0.20 \pm 0.70$ | $0.10 \pm 1.15$ | $-0.30 \pm 1.00$ | $0.05 \pm 1.70$ | $0.00 \pm 2.40$ |
| | TOTAL | $3.20 \pm 5.10$ | $10.60 \pm 3.55$ | $-2.00 \pm 3.05$ | $-2.25 \pm 2.40$ | $9.50 \pm 7.35$ |
| $SiO_4H_4$ | SW | $-0.40 \pm 0.10$ | $10.10 \pm 2.50$ | $5.75 \pm 2.50$ | $-1.80 \pm 0.70$ | $13.70 \pm 3.60$ |
| | CW | $15.80 \pm 6.35$ | $38.40 \pm 8.10$ | $6.00 \pm 5.60$ | $-2.90 \pm 3.10$ | $72.80 \pm 12.10$ |
| | IW | $18.60 \pm 31.10$ | $36.90 \pm 42.80$ | $-43.80 \pm 30.75$ | $-23.50 \pm 40.00$ | $-11.80 \pm 73.10$ |
| | DW | $3.05 \pm 12.65$ | $1.95 \pm 20.50$ | $-5.30 \pm 14.90$ | $0.80 \pm 24.80$ | $0.50 \pm 37.60$ |
| | TOTAL | $37.20 \pm 58.90$ | $87.40 \pm 29.40$ | $-37.40 \pm 55.90$ | $-27.40 \pm 29.80$ | $59.80 \pm 91.30$ |



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
