# Peer review of "Cape Verde Frontal Zone in summer 2017: lateral transports of mass, dissolved oxygen and inorganic nutrients"

_Ocean Science, 2020_

## Referee Comment (RC1) · Anonymous Referee #1 · 11 Dec 2020

The manuscript presents a novel data set from a cruise encircling the Cape Verde frontal system. The field data is complemented with climatological and numerical data. The data is well represented so that the manuscript becomes a useful description of the hydrographic conditions in a region crossed by a complex frontal system. Standard calculations on water and property transports are also incorporated.

The above is fine but the data has potential for much more. In particular, I miss a more exhaustive analysis and discussion of the results. Some suggestions follow:

1) Can the imbalances be justified in terms of the evolving velocity fields? The authors could compare the numerical and field transports for the times each individual section

was done (four sections, each carried out in about five days). They could also consider a mean numerical field during the entire period (25 days) and compare it with the combined realizations of the individual sections. Finally, they could use the numerical data to see if, at each time, the transports are balanced.

A different alternative would be to use the numerical data together with water-tracking software, such as available in https://oceanparcels.org/. That would allow the authors to differentiate between Eulerian and Lagrangian streamlines, as constructed with the numerical data.

2) The field data should be viewed as an opportunity to validate and identify the limitations of the numerical data. Possibly, numerical data does well near the surface but this may not be so at the subsurface where much less data is assimilated.

3) I would also suggest separating the region in two different domains, split by the frontal system. This would require first to set some criterion to define the position of the front, a criterion that may change with depth. Then the authors could check if properties are balanced for the tropical and subtropical domains.

4) Have the authors explored whether the imbalances in oxygen and inorgnanic nutrients are consistent? For example, if a deficit of oxygen is accompanied by a surplus of inorganic nutrients then the likely implication is remineralization. In my opinion, splitting the box in two regions would facilitate a joint interpretation of the imbalances in inorganic nutrients and oxygen.

5) Finally, I would recommend the authors to have the manuscript revised by a native English speaker. In particular, the manuscript is sometimes redundant and verbous. I also suggest to include fewer values throughout the text (tables already serve this purpose) and simply mention the most distinctive and peculiar features.

Some specific comments follow:

- I understand the convention of negative/positive values for water mass and water

properties entering/leaving the domain but I would avoid saying that e.g. "-3.2 Sv entered through the north", simply say that "3.2 Sv entered through the north".

- page 4, line 1: I believe data is not gathered every 1 dbar, this is a result of the program that interpolates the gathered data.

- page 4: have the authors explored if there is Argo data available for the region at the time of their cruise?

- page 5: could you use the temperature XBT data, aided by T-S relations, to obtain an improved resolution of the salinity fields? Would it be possible to do something similar with the inorganic nutrients and dissolved oxygen fields?

- page 5: do not use lower-case t for the temperature, rather use capital T.

- page 10, line 23: I can see a MW signal only in one station; possibly a meddy in the northern section?

- Figure A1: indicate the bathymetric contours.

- Figures A9 and A10: include a curve with the net values.

- Figures A11 and A12: the figure caption is unclear: you have scattered plots of two different variables among NO3, PO4 and dissolved oxygen. The colour bar is simultaneous for depth and neutral density, which cannot be right; it may be an approximate colour bar but you need to indicate so.

- Figure A14 and tables A1, A2: I suggest you show the transports as separated by the frontal system.

---

## Author Comment (AC1) · 31 Dec 2020

**We appreciate the comments and suggestions provided by the reviewer. We have tried to follow them to produce an improved version of the manuscript. A detailed reply is given below.**

1) **Can the imbalances be justified in terms of the evolving velocity fields? The authors could compare the numerical and field transports for the times each individual was done (four sections, each carried out in about five days). They could also consider a mean numerical field during the entire period (25 days) and compare it with the combined realizations of the individual sections. Finally, they could use the numerical data to see if, at each time, the transports are balanced.**

   We have proceeded as suggested by the reviewer, trying to evaluate with numerical model outputs if we might have had a problem of synopticity. On the one hand, we have estimated the imbalance for numerical mass transports in a given day selected in the middle of the cruise and found that the imbalance is as low as only 0.2 Sv (Fig. 1). This result indicates that the numerical model is conditioned to balance mass transports in every time step.

   On the other hand, we have produced a synthetic cruise, where we extract the model profiles in the same dates and locations of the cruise, hence introducing a time dependency in the dataset. In this second case, the imbalance is ten times larger, 2 Sv. This result indicates that the time dependency has added some noise in the velocity fields, losing some synopticity (Fig. 2). This imbalance is rather similar to that obtained from the inverse model, suggesting that indeed we might have had a problem of synopticity.

   The next version of the manuscript will contain a section to discuss with more detail this issue with the synopticity.

[Figure]

*Figure 1: Accumulated transports for SW, CW, IW and DW estimated with GLORYS on July 26.*

[Figure]

*Figure2: Accumulated transports for SW, CW, IW and DW estimated with GLORYS in the specific days in which they were performed.*

[Figure]

*Figure 3: Accumulated transports for SW, CW, IW and DW estimated with in situ data.*

**A different alternative would be to use the numerical data together with water-tracking software, such as available in https://oceanparcels.org/. That would allow the authors to differentiate between Eulerian and Lagrangian streamlines, as constructed with the numerical data.**

We consider that with the first approach we have addressed the main concern raised by the reviewer.

**2) The field data should be viewed as an opportunity to validate and identify the limitations of the numerical data. Possibly, numerical data does well near the surface but this may not be so at the subsurface where much less data is assimilated.**

The accumulated transport produced with GLORYS does not resemble any of the transports estimated with in situ data, neither at the near surface nor below (please, see figures in the previous answer). However, GLORYS recreates quite accurately the position of the front with the isohaline $S_p=36$ ($S_A=36.15$ g/kg) at 150-155 m which is detected between stations 22-23 and 32-33 in transects S and E (Figure 4).

The next version of the manuscript will present these limitations of the numerical model in this domain.

[Figure]

*Figure 4: Salinity and geostrophy fields of GLORYS with in situ values superimposed at 155 m. The position of the CVF is also indicated.*

**3) I would also suggest separating the region in two different domains, split by the frontal system. This would require first to set some criterion to define the position of the front, a criterion that may change with depth. Then the authors could check if properties are balanced for the tropical and subtropical domains.**

We really appreciate this comment. The front has historically been defined at only one depth, 150 m, where the isohaline must be 36 to identify the front. Here we have followed the suggestion provided by the reviewer to produce a method to estimate the front distribution with depth. Hence, we have taken two climatological profiles, one fully within NACW and the second one within SACW (Fig. 5a). Those climatological profiles provide a relationship between salinity, temperature, and depth. Based on those profiles, we have checked that the average salinity between those two profiles at 150 m is 36; in other words, at 150 m the front is located in a salinity where NACW and SACW contribute with 50%. We have followed the same reasoning at standard depths from 100 to 600 m depth, obtaining the salinity that would define the front location for every

depth (Figure 5b). Finally, we have performed a linear, quadratic, and cubic fit between depth and salinity, so for a given depth we can estimate the salinity that would define the front location.

Based on the quadratic fit, we have been able to depict the front location with depth, a result that have never been produced before (Fig. 6). This distribution of the front is useful to estimate the property balances in the tropical and subtropical sides of the front. Therefore, the front is considered in the three layers of CW separating the subtropical and tropical zone in which it will try to analyze the balance of properties, in new figure and table.

[Figure]

*Figure 5: a) Map with the two selected WOA stations in NACW (red) and SACW (blue) domains. b) TS diagram with the average salinity for each depth (in the range 100-650m) in black dots between the profiles of northern and southern WOA stations. c) Linear, quadratic and cubic fits for depth versus salinity.*

[Figure]

*Figure 6: Location of the front at the isohalines 36.07, 35.88, 35.67, 35.43, 35.31, 35.2 and 35.08, corresponding to average depths of 119, 190, 260, 365, 469, 584 and 698 m equivalent to 26.46, 26.63, 26.85, 26.98, 27.162, 27.28 and 27.40 kg/m³.*

4) **Have the authors explored whether the imbalances in oxygen and inorgnanic nutrients are consistent? For example, if a deficit of oxygen is accompanied by a surplus of inorganic nutrients then the likely implication is remineralization. In my opinion, splitting the box in two regions would facilitate a joint interpretation of the imbalances in inorganic nutrients and oxygen.**

This domain is particularly sensible to the suspended particle content, being a significant element in the carbon cycle (Bory et al., 2001). That is why it is not clear whether the discussion about remineralization, production and respiration could be made like in any other area of the open ocean. In addition, it would be convenient to have collected the particulate and dissolved parts of both organic and inorganic nutrients (at least C or N), which are not available in our case. The next version of the manuscript will include a discussion about the relationship between remineralization, respiration, and production processes, in terms of the contents of nutrients and oxygen.

*Bory, A., Jeandel, C., Leblond, N., Vangriesheim, A., Khripounoff, A., Beaufort, L., ... & Buat-Ménard, P. (2001). Downward particle fluxes within different productivity regimes off the Mauritanian upwelling zone (EUMELI program). Deep Sea Research Part I: Oceanographic Research Papers, 48(10), 2251-2282.

5) **Finally, I would recommend the authors to have the manuscript revised by a native English speaker. In particular, the manuscript is sometimes redundant and verbous. I also suggest to include fewer values throughout the text (tables already serve this purpose) and simply mention the most distinctive and peculiar features.**

We will make the text lighter including only relevant results such as maximum and minimum values.

**The repply to comments:**

1. **I understand the convention of negative/positive values for water mass and water properties entering/leaving the domain but I would avoid saying that e.g. "-3.2 Sv entered through the north", simply say that "3.2 Sv entered through the north"**

   We agree with the reviewer and they have been changed.

2. **page 4, line 1: I believe data is not gathered every 1 dbar, this is a result of the program that interpolates the gathered data**

   The text is changed as: "…more than 2000 m depth and processed with a vertical resolution of 1 dbar".

3. **page 4: have the authors explored if there is Argo data available for the region at the time of their cruise?**

We have now checked for Argo data in our domain and have found 7 profilers that provided 20 profiles (Figure 7). They have not been considered for the analyses in the current version of the manuscript.

[Figure]

*Figure 7: Argo profilers in the domain during FLUXES-I cruise.*

4. **page 5: could you use the temperature XBT data, aided by T-S relations, to obtain an improved resolution of the salinity fields? Would it be possible to do something similar with the inorganic nutrients and dissolved oxygen fields?**

DIVA was the interpolation method used along this manuscript because it provides estimates about the errors made in all the interpolations. DIVA has the advantage that it estimates an error field for each interpolated variable considering the distance between the interpolated and observation positions and the variability of the interpolated field. In order to check DIVA's performance, a station was randomly removed in each transect and the interpolated profile was compared with the real one. This was done for temperature and salinity with relative errors* below 10% in all the water column. For example, the maximum differences observed mainly in the shallowest depths between the original and interpolated profile in temperature were less than 0.5ºC in all the transects (Figure 8). Therefore, we considered this interpolation method valid for the rest of variables.

[Figure]

*Figure 8: Real (blue) and interpolated (red) profiles of temperature in 4 random stations at the northern, western, southern, and eastern transects with the relative error profiles indicated in black.*

*Relative_error = abs(interpolated value - real value)/ real value *100;

5. **page 5: do not use lower-case t for the temperature, rather use capital T**.

   It is changed throughout this paragraph on page 5.

6. **- page 10, line 23: I can see a MW signal only in one station; possibly a meddy in the northern section?**

   We have changed the text like this: "At IW levels a second latitudinal transition is observed between AAIW and MW from south to north (Zenk et al., 1991). In these layers, AAIW is clearly the dominant water mass with an exception at the northern area where a very diluted MW reduces the presence of AAIW."

7. **- Figure A1: indicate the bathymetric contours.**

   It has been done (Figure 9).

[Figure]

*Figure 9: Fluxes-I cruise stations with the bathymetric contour values.*

8. **Figures A9 and A10: include a curve with the net values.**

   It has been done.

9. **Figures A11 and A12: the figure caption is unclear: you have scattered plots of two different variables among NO3, PO4 and dissolved oxygen. The color bar is simultaneous for depth and neutral density, which cannot be right; it may be an approximate color bar but you need to indicate so.**

   We agree with the reviewer. Actually, the color bar presents the neutral density and the equivalent average depth. It has been changed in the figures.

10. **- Figure A14 and tables A1, A2: I suggest you show the transports as separated by the frontal system.**

    We will incorporate a new figure and table for CW layers where the front is detected as recommended by the reviewer. We will also add a section to the paper to describe the procedure followed to estimate the front location with depth.

---

## Author Comment (AC2) · 31 Dec 2020

My apologies, the reply is in the supplement file.
* * *

---

## Referee Comment (RC2) · Anonymous Referee #2 · 5 Jan 2021

This paper presents the analysis of a hydrological dataset in the Cape Verde Basin. The analysis differentiates contributions from the North from those from the South in the context of the Cape Verde Frontal Zone. This dataset and its analysis is worthy of publication, but the current limitations of the analysis mean that the conclusions are not robust enough. I have identified the major problems below. I sometimes had difficulty understanding what the authors had done or meant, so it is possible that some of my remarks are related only to a misinterpretation of the text.

The use of an inverse method to estimate volume, oxygen and nutrient transports from in situ data based on geostrophic balance and conservation equations (volume, tracers) is now a well-established approach in oceanography. Several formalisms exist, but they all amount to solving a least-squares problem whose solution will depend on errors in the observations and conservation equations used. Statistical tools exist to verify that the solution obtained is consistent and does not violate the assumptions made a priori. The application of the inverse method in this paper suffers from various limitations:

1. The joint use of an in situ data set collected in a few weeks and WOA climatology. The authors argue that the northern section and the eastern section have a strong asynopticity to build a box with hydrographic stations on the continental slope from WOA. Isn't this also a great asynopticity that is difficult to defend? WOA strongly smoothes the structures and it is difficult to believe that the boundary currents are restituted in the same way in WOA as they would have been with hydrographic data made at the same time as the section.

2. Searching for an annual-mean like solution when there is strong seasonal variability in the region.

3. I don't understand why the salinity conservation equation has a freshwater forcing term. Salt is conserved without a forcing source.

4. It is not correct to use the error on the velocities to calculate the error on the dynamic equations. The velocity error is taken into account in the velocity term of the cost function. It should not be taken into account twice.

5. In general, there should be a table that summarizes the different parameters of the inverse model and presents the different errors used, and the a posteriori errors as well.

6. What is missing is an a posteriori analysis of the solution to verify that the a priori hypotheses are satisfied. GLORYS could also be compared to the reference level velocities estimated from the inversion. If there is good agreement, this validates both

the inversion and the use of GLORYS velocities on section E. If there is not good agreement it will invalidate the method.

7. The data do not resolve the sub-mesoscale which, as indicated in the introduction, plays an important role in property transfers in this region.

In several places in the text you repeat information already given in the figure captions (see for example page 7, lines 22 to 27). This makes the text unnecessarily heavy.

Add the locations of the CVFZ in figures A2, A4, A5, A7.

Figure A8 and associated discussion. I can see an anticyclonic eddy between station 3 and 6 on figure A8 but I can't see a cyclonic eddy between station 4 and 6 on the same figure. Please clarify. Add a few station numbers on Figure A8, it will help to follow the discussion.

The term filament (page 9, line 12) is misleading as filament dynamics in the presence of a mixed layer is not adequately described by the classical thermal wind balance and thus cannot be resolved by altimetry.

Figures A9 and A10 are difficult to interpret, especially when the results of the different sections have to be connected to each other. I suggest to use Figure A14 and similar figures for biogeochemical transports to better convey the message in the discussion of these transports.

Figures A9, A10 and A14: it is not clear if section N and S include the transports from the slope and shelf regions where WOA hydrography was used.

Figure A14 shows very well that mass balance is far from being satisfied. This become a very serious problem when interpreting the biogeochemical tracer transport imbalances in terms of accumulation/consumption of the tracer while the primary reason for this imbalance is that the mass is not conserved. It is also a problem for the mass balance leading to sentences like the one in page 12 line 7: a significant input of -1 +/-1.3 Sv. I would not say it is significant. The direction is not robust.

Concerning the mass balance, I find it hard to believe that the inclusion of section E in the inversion would have created a problem of synopticity such that the mass would have been less well conserved than with the current solution.

Figures A11 and A12. I'm not quite sure what these figures and the discussion that goes with them add to the discussion of figures A2 to A7 (most of the discussion is spent arguing that this data set agrees with previous ones). At a minimum, the figures and the two discussions should be grouped together.

---

## Referee Comment (RC3) · Anonymous Referee #3 · 12 Jan 2021

Review of os-2020-98 "Lateral transports of mass, inorganic nutrients and dissolved oxygen in the Cape Verde Frontal Zone in summer 2017" Burgoa et al.

The manuscript describes the distribution of hydrographic, nutrients and oxygen concentrations along the sides of a large box that covers the upwelling area and CVFZ in the cape verde basin, along the Western Sahara and Mauritanian coasts during the summer of 2017 ($\sim$26 days).

Combining full depth CTD, XBT profiles and water samples the authors describe the distribution of water masses along the ships' track. The XBT data are used to increase the resolution of the temperature data in all transects and "optimal" interpolation parameters are generated through the use of DIVA interpolation package. These are later used to interpolate the more sparse dataset of salinity (and hence density for mass transport estimates) and biochemical variables. Through the application of an inverse transport model the authors estimate mass and biochemical fluxes across the transects using observed (and climatological) geostrophic velocities as first guesses in an 8 layer distribution (defined approximately to separate the contributions by the major water masses in a similar manner to previous estimates for the region).

The main conclusions from this exhaustive exercise are that the distribution, characteristics and transports by each water mass are comparable to previous estimates already published for the region (and referenced in this manuscript). The detailed estimates per isoneutral layer are useful to compare with both climatological estimates and modelling products and I see value in the publication of these results.

General comments

The long time taken to complete the sampling introduces unavoidable aliases in the distribution of the variables and geostrophic velocities and this is reflected in the large errors associated with both biochemical and mass transports. The approach has been to use the annual mean estimate of the velocity variance at the reference level from GLORYS. Would it be better to use global model outputs instead that would have a higher resolution (more comparable to the resolution of the observations?). As for the validation, it would be very useful to compare the geostrophic transport estimates with the ship's ADCP data, at least for the SW and CW layers.

I would consider moving figures A8 and A13 earlier in the results section to provide a seasonal context to the observations. I would also welcome a brief description of the conditions in 2017 with respect to the other years to put the values into a wider context (maybe this can be done in the discussion instead).

I am curious to see the impact of using DIVA vs a standard krigging interpolation scheme. Would it be possible to include an example in the supplementary materials

section? Do you think this would have an impact on the transport estimates?

Is there any reason why the water mass content was not quantified through the Optimal Multiparameter Method? (OMP) (i.e. Zhou, P., Song, X., Yuan, Y., Cao, X., Wang, W., Chi, L., & Yu, Z. (2018). Water mass analysis of the East China Sea and interannual variation of Kuroshio Subsurface Water intrusion through an Optimum Multiparameter method. Journal of Geophysical Research: Oceans, 123, 3723‐ 3738. https://doi.org/10.1029/2018JC013882)

Grammar/English There are some instances in which the text is difficult to follow due to complex descriptions and syntax that could benefit from the input of an english speaker. I know how difficult it is to summarise such a large dataset but adding every detail dilutes the main messages that one would like to convey. It is important that the descriptions be kept short with simple sentences to help the reader. This can be aid by anotating the figures (i.e. N, W , S and E in the title sections of the transect figures, depth labels in A1, special features/water masses in figs A5-6, major current names in A7, features described in the text in A8 such as CVFZ etc.. )

The discussion and conclusions section would need to be revise in detail to make it easier to follow.

Detailed comments

P1L15 hinder (instead of hinders)

P2L1 Being a permanent upwelling area the CUF is always present (intensity and location might change). revise

P2L6 try and mimic SACW description (i.e. add location of where it forms)?

P2L15 IS modifiED

P2L14-21 Revise paragraph

P3L13-19 Include what sets the present manuscript apart from the papers cited.

[Figure]

P4L22 - Add who provided the wind data (i.e. url or data provider)

P5L25.. I am afraid I couldn't understand this paragraph and what it meant. Could you point in A1 what stations were in fact climatological nodes?

P7L5. This is an area where diapycnal mixing is significant, specially in the CW layers and in the confluence of NACW and SACW as here. Dismissing them from the inverse model might be another reason for the large errors in the estimates and the lack of mass conservation in the results. (Peña-Izquierdo, J., . On the circulation of the North Atlantic shadow zone 150., Peña‐Izquierdo, J., Sebille, E. van, Pelegrí, J.L., Sprintall, J., Mason, E., Llanillo, P.J., Machín, F., 2015. Water mass pathways to the North Atlantic oxygen minimum zone. Journal of Geophysical Research: Oceans 120, 3350–3372. https://doi.org/10.1002/2014JC010557)

P7L17 definitions for instead of "relationship between"

P7L29 included instead of "grouped"

P8L12-27 Revise text to make it clearer.

P8 I think this is where having the OMP results might help the description. This can complement the information in A11 and A12 too.

Clarify terminology when refering to eddy, meanders and filaments. Sometimes, their use in the text is confusing.

P9L18-19 revise sentence.

P24-25 Label of A11 and A12. Second NO3 should be PO4.

---

## Author Comment (AC3) · 8 Feb 2021

The comments and suggestions provided by the reviewer are very much appreciated. We have tried to follow them to produce an improved version of the manuscript. A detailed reply (in bold) to each specific query (in light) is given below.

1. The joint use of an in situ data set collected in a few weeks and WOA climatology. The authors argue that the northern section and the eastern section have a strong asynopticity to build a box with hydrographic stations on the continental slope from WOA. Isn't this also a great asynopticity that is difficult to defend? WOA strongly smoothes the structures and it is difficult to believe that the boundary currents are restituted in the same way in WOA as they would have been with hydrographic data made at the same time as the section.

The objective of the procedure followed is to produce a dataset that allows us to consider mass conservation within a closed volume minimizing the impact of asynopticity. As pointed out by the reviewer, WOA provides smooth data which finally helps to produce an average mass transport close to the continental slope. Such a smooth and average nature of this dataset makes them particularly suitable in this context to avoid adding any imbalances to the dataset.

2. Searching for an annual-mean like solution when there is strong seasonal variability in the region.

We agree with the reviewer and we have modified the manuscript accordingly. The numerical model GLORYS is now employed to develop a climatological summer velocity field from 25 years of data. This climatological summer velocity field is used as the a priori reference level velocity in the four transects. In addition, the seasonal variability of velocity in the eight layers is estimated from the summer months covered by those 25 years of GLORYS.

3. I don't understand why the salinity conservation equation has a freshwater forcing term. Salt is conserved without a forcing source.

These equations are added to the inverse model as anomaly equations. The fresh water flux does not affect the salt transport but it does affect the salt anomaly transport. In the definition of salt anomaly transport,  $T'_{S}$ , given by:

$$T'_{S} = T_{S} - \overline{S_{c}} \times T_{M}$$

 $T_S$  is the salt transport,  $\overline{S_c}$  is the mean salinity in a layer and  $T_M$  is the mass transport which includes the whole mass even the freshwater. So once the subtraction is performed, the freshwater term appears in the salt anomaly equation (Ganachaud, 2003).

\*Ganachaud, A. (2003). Error budget of inverse box models: The North Atlantic. *Journal of Atmospheric and Oceanic Technology*, 20(11), 1641-1655.

4. It is not correct to use the error on the velocities to calculate the error on the dynamic equations. The velocity error is taken into account in the velocity term of the cost function. It should not be taken into account twice.

We disagree with the reviewer. These procedure has been applied previously with satisfactory results (Machín et al., 2006; Burgoa et al., 2020). In particular, Ganachaud (2003)\* made an analysis of these errors concluding that mass transports and their imbalances are largely related to the error of the velocity, since the sea water density and vertical areas to estimate lateral transports are known more accurately. Hence, the main source of uncertainty in the equations comes from the error in the velocity.

\*Ganachaud, A. (2003). Error budget of inverse box models: The North Atlantic. *Journal of Atmospheric and Oceanic Technology*, 20(11), 1641-1655.

5. In general, there should be a table that summarizes the different parameters of the inverse model and presents the different errors used, and the a posteriori errors as well.

We agree with the reviewer and have modified the manuscript accordingly. On the one hand, the values for a priori uncertainties are included in a new table on "*Characteristics and constrains*" of the inverse model. On the other hand, in a new sub-section about the solution of the inverse model, a new figure (right on Fig. 4) shows that the full-depth integrated error is around 1 Sv for the mass transports extracted from the inverse model and the text has been modified to state that the errors in the reference velocities are below 0.025 m s-1.

6. What is missing is an posteriori analysis of the solution to verify that the a priori hypotheses are satisfied. GLORYS could also be compared to the reference level velocities estimated from the inversion. If there is good agreement, this validates both the inversion and the use of GLORYS velocities on section E. If there is not good agreement it will invalidate the method.

We have followed the comment provided by the reviewer and instead of comparing the velocities at the reference level where the differences between the velocities estimated with the inversion (of the order of 10-3) and those by the numerical model GLORYS are negligible, we have performed the comparison at the sea surface, where velocities are higher and also their potential impact on the integrated transports in the water column. In addition, we have also used the geostrophy derived from altimetry as a third independent element for validation. The comparison is performed in terms of the accumulated transports in the first layer. As shown in Figure 1 in this document, the accumulated transports estimated with the inversion including GLORYS' velocities as the reference velocities have the same behaviour as the accumulated transports using both the geostrophy derived from altimetry and from GLORYS, being the final difference among the three methods in the order of 1 Sv. That result supports using this methodology in the present work.

Figure 1: Accumulated transports in the first layer estimated along transects N, W and S (without WOA stations) with altimetry's derived geostrophy (red line), inversion (with GLORYS' as reference velocities, black line) and GLORYS' field (blue line).

7. The data do not resolve the sub-mesoscale which, as indicated in the introduction, plays an important role in property transfers in this region.

We agree with the reviewer and we discuss it now in the final section. With this methodology it is not possible to distinguish the percentage of the transport which is due to the meso and sub-meso-scale (probably part of the error is due to these processes). Samplings with lower resolutions should be carried out to study in detail the role of the meso- and submesoscale in the transfer of properties as the one developed by Hosegood et al. (2017) in the study area.

\*Hosegood, P. J., Nightingale, P. D., Rees, A. P., Widdicombe, C. E., Woodward, E. M. S., Clark, D. R., & Torres, R. J. (2017). Nutrient pumping by submesoscale circulations in the mauritanian upwelling system. Progress in Oceanography, 159, 223-236.

In several places in the text you repeat information already given in the figure captions (see for example page 7, lines 22 to 27). This makes the text unnecessarily heavy.

**We have reviewed these redundancies to make the manuscript lighter.**

Add the locations of the CVFZ in figures A2, A4, A5, A7.

We include a new figure (Fig. 2 in this document) with the vertical location of the front on a map and also two vertical sections produced by an OMP analysis. We have not added it to the rest of figures to not overload the images.

Figure 2: a) Location of the front at the isohalines 36.07, 35.88, 35.67, 35.43, 35.31, 35.2 and 35.08, corresponding to average depths of 119, 190, 260, 365, 469, 584 and 698 m equivalent to 26.46, 26.63, 26.85, 26.98, 27.162, 27.28 and 27.40 kgm-3. Vertical sections of the three layers of CW with the percentages of NACW (b) and SACW (c) and the front location indicated by pink lines. The 4 transects are separated by three vertical gray dashed lines located at stations number 12, 19 and 28. Three layers are also separated by two horizontal gray dashed lines.

Figure A8 and associated discussion. I can see an anticyclonic eddy between station 3 and 6 on figure A8 (3 y 5) but I can't see a cyclonic eddy between station 4 and 6 on the same figure (3 y 5). Please clarify. Add a few station numbers on Figure A8, it will help to follow the discussion.

Some stations numbers have been included in Figure A8 (it is updated below in Fig. 3 of this document). In the northern transect, the first small anticyclonic eddy is centred at station 4. Next to it, there is a larger cyclonic eddy between stations 4 and 7 with its centre between

stations 5 and 6. This description is made simultaneously observing this figure and the vertical section of velocities.

---

## Author Comment (AC4) · 8 Feb 2021

We appreciate the suggestions and comments provided by the reviewer. We have tried to Include them in the revised version of the manuscript.

**General comments**

The long time taken to complete the sampling introduces unavoidable aliases in the distribution of the variables and geostrophic velocities and this is reflected in the large errors associated with both biochemical and mass transports. The approach has been to use the annual mean estimate of the velocity variance at the reference level from GLORYS. Would it be better to use global model outputs instead that would have a higher resolution (more comparable to the resolution of the observations?).

In the revised version of the manuscript, we have used the climatological variability obtained for the summer months obtained from 25 years of GLORYS, which is a global model reanalysis. We have not used a global prediction model because the outputs are worse than those of the reanalysis. On the other hand, global models with higher resolution are not available for our area.

As for the validation, it would be very useful to compare the geostrophic transport estimates with the ship's ADCP data, at least for the SW and CW layers.

Figure 1: Three examples of two subsequent across-section? velocity profiles from sADCP (dark and light blue lines) together with the mean between them (pink line) and the moving mean of this last mean (dashed black line). The geostrophic profiles obtained with the density field in the middle position between two sADCP profiles are shown for comparison.

An attempt was made to perform this validation. The velocity profiles obtained from the in situ density field and those measured by the sADCP had uneven shapes in many stations (Figure 1 of this document), so it was not feasible to reference the reference level geostrophic velocity to the sADCP samplings, as suggested by the reviewer. In addition, we believe that the sADCP was not correctly calibrated prior to sampling, so they have not been used in this work.

I would consider moving figures A8 and A13 earlier in the results section to provide a seasonal context to the observations. I would also welcome a brief description of the conditions in 2017 with respect to the other years to put the values into a wider context (maybe this can be done in the discussion instead).

**We appreciate the suggestion; we have modified the discussion section according to this comment.**

I am curious to see the impact of using DIVA vs a standard krigging interpolation scheme. Would it be possible to include an example in the supplementary materials section? Do you think this would have an impact on the transport estimates? Performing the comparison suggested by the reviewer is likely beyond the goal of this manuscript. We have now added references to specific works where those comparisons are performed (Barth et al., 2010; Troupin et al., 2012; Beckers et al., 2014). The interpolations are made with DIVA because it is an objective mapping which computes and gives us the error made in all the interpolations. In this way the precision of the method is checked. In the updated text, it is pointed out that the interpolations with an error greater than 10% are not considered in the subsequent analysis.

\*Barth, A., Alvera-Azcárate, A., Troupin, C., Ouberdous, M., & Beckers, J. M. (2010). A web interface for griding arbitrarily distributed in situ data based on Data-Interpolating Variational Analysis (DIVA). Advances in Geosciences, 28, 29-37.

\*Troupin, C., Barth, A., Sirjacobs, D., Ouberdous, M., Brankart, J. M., Brasseur, P., ... & Beckers, J. M. (2012). Generation of analysis and consistent error fields using the Data Interpolating Variational Analysis (DIVA). Ocean Modelling, 52, 90-101.

\*Beckers, J. M., Barth, A., Troupin, C., & Alvera-Azcárate, A. (2014). Approximate and efficient methods to assess error fields in spatial gridding with data interpolating variational analysis (DIVA). Journal of Atmospheric and Oceanic Technology, 31(2), 515-530.

The interpolation results have been verified with the mass transports. Figure 2 of this document shows the differences between estimating the accumulated mass transport using only the CTD stations (along the transects N, W and S) or also including the reconstructed stations with interpolated salinity data and in-situ temperature from XBTs.

Figure 2: Accumulated mass transports per water types layers along transects N, W and S employing only CTD stations (up) and CTD plus interpolated stations (down).

**The accumulated mass transports which include the interpolated stations are noisier (down in Fig. 2), with a similar shape to those estimated only with CTD.**

Is there any reason why the water mass content was not quantified through the Optimal Multiparameter Method? (OMP) (i.e. Zhou, P., Song, X., Yuan, Y., Cao, X., Wang, W., Chi, L., & Yu, Z. (2018). Water mass analysis of the East China Sea and interannual variation of Kuroshio Subsurface Water intrusion through an Optimum Multiparameter method. Journal of Geophysical Research: Oceans, 123, 3723â °A °R 3738. https://doi.org/10.1029/2018JC013882)

We have included an OMP analysis in the updated version mainly to quantify the central waters NACW and SACW. In this way, we could assess the vertical and horizontal location of the front between both water masses. The front location estimated with the OMP analysis compares well with that obtained with a new methodology provided in the updated version of this manuscript, where WOA climatological data are used to extend vertically the classical definition of the Cape Verde Front (Figure 3 of this document).

---

## Author Comment (AC5) · 8 Feb 2021

**We appreciate the suggestions and comments provided by the reviewer. We have tried to Include them in the revised version of the manuscript.**

General comments

The long time taken to complete the sampling introduces unavoidable aliases in the distribution of the variables and geostrophic velocities and this is reflected in the large errors associated with both biochemical and mass transports. The approach has been to use the annual mean estimate of the velocity variance at the reference level from GLORYS. Would it be better to use global model outputs instead that would have a higher resolution (more comparable to the resolution of the observations?).

**In the revised version of the manuscript, we have used the climatological variability obtained for the summer months obtained from 25 years of GLORYS, which is a global model reanalysis. We have not used a global prediction model because the outputs are worse than those of the reanalysis. On the other hand, global models with higher resolution are not available for our area.**

As for the validation, it would be very useful to compare the geostrophic transport estimates with the ship's ADCP data, at least for the SW and CW layers.

[Figure]

*Figure 1: Three examples of two subsequent across-section? velocity profiles from sADCP (dark and light blue lines) together with the mean between them (pink line) and the moving mean of this last mean (dashed black line). The geostrophic profiles obtained with the density field in the middle position between two sADCP profiles are shown for comparison.*

**An attempt was made to perform this validation. The velocity profiles obtained from the in situ density field and those measured by the sADCP had uneven shapes in many stations (Figure 1 of this document), so it was not feasible to reference the reference level geostrophic velocity to the sADCP samplings, as suggested by the reviewer. In addition, we believe that the sADCP was not correctly calibrated prior to sampling, so they have not been used in this work.**

I would consider moving figures A8 and A13 earlier in the results section to provide a seasonal context to the observations. I would also welcome a brief description of the conditions in 2017 with respect to the other years to put the values into a wider context (maybe this can be done in the discussion instead).

**We appreciate the suggestion; we have modified the discussion section according to this comment.**

I am curious to see the impact of using DIVA vs a standard krigging interpolation scheme. Would it be possible to include an example in the supplementary materials section? Do you think this would have an impact on the transport estimates?

Performing the comparison suggested by the reviewer is likely beyond the goal of this manuscript. We have now added references to specific works where those comparisons are performed (Barth et al., 2010; Troupin et al., 2012; Beckers et al., 2014). The interpolations are made with DIVA because it is an objective mapping which computes and gives us the error made in all the interpolations. In this way the precision of the method is checked. In the updated text, it is pointed out that the interpolations with an error greater than 10% are not considered in the subsequent analysis.

*Barth, A., Alvera-Azcárate, A., Troupin, C., Ouberdous, M., & Beckers, J. M. (2010). A web interface for griding arbitrarily distributed in situ data based on Data-Interpolating Variational Analysis (DIVA). Advances in Geosciences, 28, 29-37.

*Troupin, C., Barth, A., Sirjacobs, D., Ouberdous, M., Brankart, J. M., Brasseur, P., ... & Beckers, J. M. (2012). Generation of analysis and consistent error fields using the Data Interpolating Variational Analysis (DIVA). Ocean Modelling, 52, 90-101.

*Beckers, J. M., Barth, A., Troupin, C., & Alvera-Azcárate, A. (2014). Approximate and efficient methods to assess error fields in spatial gridding with data interpolating variational analysis (DIVA). Journal of Atmospheric and Oceanic Technology, 31(2), 515-530.

The interpolation results have been verified with the mass transports. Figure 2 of this document shows the differences between estimating the accumulated mass transport using only the CTD stations (along the transects N, W and S) or also including the reconstructed stations with interpolated salinity data and in-situ temperature from XBTs.

[Figure]

*Figure 2: Accumulated mass transports per water types layers along transects N, W and S employing only CTD stations (up) and CTD plus interpolated stations (down).*

**The accumulated mass transports which include the interpolated stations are noisier (down in Fig. 2), with a similar shape to those estimated only with CTD.**

Is there any reason why the water mass content was not quantified through the Optimal Multiparameter Method? (OMP) (i.e. Zhou, P., Song, X., Yuan, Y., Cao, X.,Wang, W., Chi, L., & Yu, Z. (2018). Water mass analysis of the East China Sea and interannual variation of Kuroshio Subsurface Water intrusion through an Optimum Multiparameter method. Journal of Geophysical Research: Oceans, 123, 3723ăˇAˇR 3738. https://doi.org/10.1029/2018JC013882)

**We have included an OMP analysis in the updated version mainly to quantify the central waters NACW and SACW. In this way, we could assess the vertical and horizontal location of the front between both water masses. The front location estimated with the OMP analysis compares well with that obtained with a new methodology provided in the updated version of this manuscript, where WOA climatological data are used to extend vertically the classical definition of the Cape Verde Front (Figure 3 of this document).**

[Figure]

*Figure 3: a) Location of the front at the isohalines 36.07, 35.88, 35.67, 35.43, 35.31, 35.2 and 35.08, corresponding to average depths of 119, 190, 260, 365, 469, 584 and 698 m equivalent to 26.46, 26.63, 26.85, 26.98, 27.162, 27.28 and 27.40 kgm$^{-3}$. Vertical sections of the three layers of CW with the percentages of NACW (b) and SACW (c) and the front location indicated by pink lines. The 4 transects are separated by three vertical gray dashed lines located at stations number 12, 19 and 28. Three layers are also separated by two horizontal gray dashed lines.*

Grammar/English There are some instances in which the text is difficult to follow due to complex descriptions and syntax that could benefit from the input of an english speaker. I know how difficult it is to summarise such a large dataset but adding every detail dilutes the main messages that one would like to convey. It is important that the descriptions be kept short with simple sentences to help the reader. This can be aid by anotating the figures (i.e. N, W , S and E in the title sections of the transect figures, depth labels in A1, special features/water masses in figs A5-6, major current names in A7, features described in the text in A8 such as CVFZ etc.. )

**We have revised the English grammar and style of the manuscript. In addition, the updated manuscript includes relevant information in the figures to facilitate its understanding and to easily follow the text.**

The discussion and conclusions section would need to be revise in detail to make it easier to follow.

**We have thoroughly reviewed the discussion and conclusions to improve its understanding.**

Detailed comments

P1L15 hinder (instead of hinders)

**We changed it.**

P2L1 Being a permanent upwelling area the CUF is always present (intensity and location might change). Revise

**CUF exists where upwelling is permanent year-round (Benazzouz et al., 2014). In our case this happens only north of Cape Blanc. The trade winds are intense all year round between Cape Blanc and the Canary Islands, reaching Cape Vert during winter (Pelegri et Bennazouz, 2015).**

**\*Benazzouz, A., Pelegrí, J. L., Demarcq, H., Machín, F., Mason, E., Orbi, A., ... & Soumia, M. (2014). On the temporal memory of coastal upwelling off NW Africa. Journal of Geophysical Research: Oceans, 119(9), 6356-6380.**

**\*Pelegrı, J. L., & Benazzouz, A. (2015). Coastal Upwelling off North-West Africa, Oceanographic and Biological Features in the Canary Current Large Marine Ecosystem. IOC-UNESCO, Paris, 93-103.**

P2L6 try and mimic SACW description (i.e. add location of where it forms)?

**We have rephrased the sentence to: "The northern side of the CVFZ is mainly occupied by waters of different subtropical origin grouped as Eastern North Atlantic Central Water (NACW) which flows southward transported by the Canary Current (CC)".**

**In the new section where we show the contributions of NACW and SACW estimated by the OMP (Figure 3), it is pointed out that NACW is indeed composed of Madeira Mode Water (MMW) and Eastern NACW of 15 and 12 ºC.**

P2L15 IS modifiED

**We changed it.**

P2L14-21 Revise paragraph

**This paragraph deals with the detailed path followed by SACW from the southern hemisphere to the domain of interest. The reviewer suggests revising it without giving any additional details, so we have focused our action in adding references to support the content presented in the paragraph. The text is now presented as follows:**

**"South Atlantic Central Water (SACW) is the main water mass at the southern side of the CVFZ. This water mass is formed at the subtropical South Atlantic and it is modified after crossing the tropical regions (Peña-Izquierdo et al., 2015). SACW penetrates into the Cape Verde Basin via the northern branch of the North Equatorial Countercurrent reaching the African Slope as the Cabo Verde Current (CVC) (Peña-Izquierdo et al., 2015; Pelegrí et al., 2017). CVC move**

**anticlockwise around the Guinea Dome (GD) reaching to the southern part of the CVFZ (Peña-Izquierdo et al., 2015; Pelegrí et al., 2017) with a seasonal variability mainly driven by latitudinal changes in the Inter-Tropical Converge Zone (ITCZ) (Siedler et al., 1992). In summer, GD intensifies as a result of the northward penetration of ITCZ (Castellanos et al., 2015). In addition, the northward flow along the African coast intensifies due to the relaxation of trade winds at latitudes south of Cape Blanc, so Mauritanian Current and PUC can reach just south of Cape Blanc in this season (Siedler et al., 1992; Lázaro et al., 2005)."**

**\* Peña-Izquierdo, J., van Sebille, E., Pelegrí, J. L., Sprintall, J., Mason, E., Llanillo, P. J., and Machín, F.: Water mass pathways to the North Atlantic oxygen minimum zone, Journal of Geophysical Research: Oceans, 120, 3350–3372, 2015.**

**\* Pelegrí, J. L., Peña-Izquierdo, J., Machín, F., Meiners, C., and Presas-Navarro, C.: Deep-Sea Ecosystems Off Mauritania, Chapter 3, Oceanography of the Cape Verde Basin and Mauritanian Slope Waters, Springer, https://doi.org/10.1007/978-94-024-1023-5_3, http://api.elsevier.com/content/abstract/scopus_id/85035361292, 2017.**

P3L13-19 Include what sets the present manuscript apart from the papers cited.

**This manuscript aims to address the circulation patterns and the physical processes behind the distribution of $O_2$ and inorganic nutrients at the dynamically complex CVFZ, a domain where *in situ* data availability has historically been very limited. Secondly, we have extended the classical definition of the CVF to assess its location with depth, a result that has never been produced before. The CVF acts as a barrier and a source of meso- and sub-mesoscale variability, so now the transports of mass, $O_2$ and nutrients can be estimated independently on the subtropical and tropical domains, evaluating how the front affects all transports and producing an interpretation of the imbalances in $O_2$ and inorganic nutrients.**

P4L22 - Add who provided the wind data (i.e. url or data provider)

**We have updated the url direction to:**

**ftp://ftp.ifremer.fr/ifremer/cersat/products/gridded/MWF/L3/ASCAT/Daily/**

P5L25. I am afraid I couldn't understand this paragraph and what it meant. Could you point in A1 what stations were in fact climatological nodes?

**We include these stations with green dots in this figure (Figure 4 of this document).**

[Figure]

*Figure 4: CTD-rosette sampling stations (pink dots) and XBT (blue dots) during FLUXES-I cruise. There are also represented WOA stations (green dots). Time-averaged wind stress during the cruise is also represented with the inset arrow denoting the scale (shown with half of the original spatial resolution).*

P7L5. This is an area where diapycnal mixing is significant, specially in the CW layers and in the confluence of NACW and SACW as here. Dismissing them from the inverse model might be another reason for the large errors in the estimates and the lack of mass conservation in the results. (Peña-Izquierdo, J., . On the circulation of the North Atlantic shadow zone 150., Peñaâ ˇA ˇRIzquierdo, J., Sebille, E. van, Pelegrí, J.L., Sprintall, J., Mason, E., Llanillo, P.J., Machín, F., 2015. Water mass pathways to the North Atlantic oxygen minimum zone. Journal of Geophysical Research: Oceans 120,3350–3372. https://doi.org/10.1002/2014JC010557)

**The diapycnal turbulent diffusivity value, $K_s = 10^{-5}$ m² s⁻¹, estimated from in-situ data for the column between 150-600 m in our area by Martinez-Marrero et al. (2008), allows us to estimate the diapycnal mixing in the central waters domain:**

**$K_s=10^{-5}$ m² s⁻¹;**

**Z=600-150 m= 450 m;**

**Area of the rectangle described by stations (A) = height * base =540 km*770 km ≈ $4\times10^{11}$ m²**

**Mean density (ρ) = 1027 kg m⁻³**

**Diapycnal transport= $K_s * ρ * A / z = 9\times10^6$ kg s⁻¹ ≈ 0.01 Sv**

**This diapycnal transport of 0.01 Sv is small as compared to the estimated isopycnal transports. Therefore, this transport is negligible and it will not introduce major errors in the method followed.**

*Martínez-Marrero, A., Rodríguez-Santana, A., Hernández-Guerra, A., Fraile-Nuez, E., López-Laatzen, F., Vélez-Belchí, P., & Parrilla, G. (2008). Distribution of water masses and diapycnal mixing in the Cape Verde Frontal Zone. *Geophysical Research Letters*, *35*(7).

P7L17 definitions for instead of "relationship between"

**We changed it.**

P7L29 included instead of "grouped"

**We modified it.**

P8L12-27 Revise text to make it clearer.

**We have re-written it as follows:**

**The distributions of $O_2$, $NO_3$ and $PO_4$ and $SiO_4H_4$ (Figs. A6 and A7) were highly variable mainly in the CW and IW layers with a notable minimum concentration of $O_2$ (60-90 µmol kg$^{-1}$ between 100 and 800 m), two large maximum values of $NO_3$ and $PO_4$ (30-32 and 2 µmol kg$^{-1}$ respectively between 350 and 1000 m) and a deeper and less abrupt maximum value of $SiO_4H_4$ (21 µmol kg$^{-1}$ below 700 m) centred in transect S. These marked distributions were closely related to the distribution of the different water masses.**

**In transects N and W, where NACW was found, the concentrations of $O_2$ were higher than in transects S and E, where SACW was found. For instance, concentrations of $O_2$ lower than 60 µmol kg$^{-1}$ are observed at 300 m in transects S and E (Fig. A6). In contrast, the concentrations of the three inorganic nutrients in these last two transects were higher than in transects N and W at CW levels. For example, concentrations around 27-30 µmol kg$^{-1}$ of $NO_3$, 1.5-1.7 µmol kg$^{-1}$ of $PO_4$, and 7.5-9.9 µmol kg$^{-1}$ of $SiO_4H_4$ are observed at 300 m depth in transects S and E (Figs. 6 and 7).**

**Below the first layer of IW, the distribution of $O_2$ was quite uniform. With respect to inorganic nutrient distributions at IW levels, transect N had lower concentrations than the rest of transects which had a higher content of AAIW. Figures A6 and A7 show concentrations higher than 33, 2.05 and 21.4 µmol kg$^{-1}$ of $NO_3$, $PO_4$ and $SiO_4H_4$, respectively, associated to AAIW around 1000 m in transects S and E.**

**In the deepest layer, high concentrations of $O_2$ and inorganic nutrients were found. Specifically, the concentrations of $SiO_4H_4$ were the highest of the water column while the concentrations of $NO_3$ and $PO_4$ were lower than their maximum values observed between 350 and 1000 m.**

P8 I think this is where having the OMP results might help the description. This can complement the information in A11 and A12 too.

**The reviewer is right. We have discussed it in the last version of the manuscript.**

Clarify terminology when refering to eddy, meanders and filaments. Sometimes, their use in the text is confusing.

**We have changed the filament term by "intrusion". We also include some station numbers in Figure A8 to follow the text more easily.**

P9L18-19 revise sentence.

**We have revised and changed it:**

**"The velocities in transect E were the smallest ones making difficult to identify the main structures and to link them with altimetry. In addition, the high variability in this transect due to the proximity of the coast and upwelling system makes also difficult to deeply describe the estimated velocities."**

P24-25 Label of A11 and A12. Second NO3 should be PO4

**The reviewer is right. It has already been changed.**

---

## Author Response (AR1)

First of all, we would like to thank the editorial board for the extension granted in the deadline.

Secondly, all corrections and suggestions made by the editor and the three referees are greatly appreciated. They have been very important to improve the manuscript. In fact, the work has been deeply modified because results have changed and new sections, figures, and tables related to the points raised by the editor and referees are now included.

In order to minimize the imbalances, the results have been recalculated after slightly modifying the methodology and the a priori uncertainties. In this way, we consider the current results to be more robust than those provided in the previous version of the manuscript.

The uncertainties related to the a priori geostrophic dynamic have been modified and instead of taking the variances from a single year, they are calculated from summer months of 25 years of GLORYS. On the other hand, initially, the average summer climatological velocities (also estimated from summer months of 25 years of GLORYS) are included at the reference level in all the transects. Later, these velocities are modified with the inversion in transects north, west, and south.

A major rewriting of the paper has been performed, omitting redundant sentences and repeated information, reducing the values in the text which are represented in the figures and tables with the aim to facilitate the reading. In addition, the estimated uncertainties made in the applied methodologies has been included on the paper, as for example the new table that groups the a priori uncertainties of the inverse model (Tab. A1).

Two new subsections are now added to the methodology, one related to the Optimum Multiparameter method to estimate the water masses distribution and another one related to the front detection with depth (which includes a new figure, Fig. A2). Further, two new subsections are introduced in the results where the disposition of the front at different depths is presented (with a new figure, Fig. A10) and the results of mass transports estimated by the inverse model are shown (Fig. A11). In fact, the front distribution with depth is one of the most relevant results extracted from this work. It is noteworthy that the current net mass transport presents a small imbalance after the inverse method is applied. In this way, the imbalances in the oxygen and nutrient transports at surface and central levels are mainly due to biogeochemical processes beyond the physical forcing.

On the other hand, taking into account that NACW is practically Eastern NACW, NACW is renamed as ENACW throughout the document.

Two figures related to the physico-chemical characteristics of the water masses measured during the cruise (Figs. A11 and A12 in the old discussion) have been moved to the results section where they are described with the rest of the hydrology (Figs. A8 and A9).

The table where the mass transports were grouped by layers and transects (old Tab. A1) is eliminated from the updated manuscript because Figure A11b present the values in each layer. Moreover, the mass transports grouped by water masses levels is included with the transports of oxygen and inorganic nutrients in the new Table A2. On the other hand, the transports through transect E are eliminated from this table where the imbalances are estimated without them (it is not part of the current inversion box).

Figures A13 and A14 are removed from the old discussion and conclusions section. A new figure is included to describe the transports through transect E in an independent way (Fig. A15).

Moreover, the figure where the shallowest transports (in the first layer) are validated with the altimetry (old Fig. A8) is replaced with a new one where the accumulated transports estimated by altimetry, numerical model and the first layer of the inverse model are compared (Fig. A12).

Discussion and conclusions are separated into two independent sections. In the discussion the transports and imbalances of mass, oxygen and nutrients are discussed independently in both sides of the front (supported by Tab. A3).

Next, point by point, the answers to the issues and questions posed by the editor and the referees, already sent to the open discussion, are presented.

**Comments to the Author:**

This paper discusses the results from a survey of a box off the coast of NW Africa that covered the zone of interaction between the North Atlantic subtropical and tropical gyres, where NACW and SACW provide inputs that are distinct in terms of their oxygen and nutrient concentrations across a frontal zone. The paper seems well constructed, although I had some difficulty in following their arguments in the discussion section without constantly referring back to their definitions of the density layer boundaries. I suggest they add these quantities to the text where needed as this would help the reader decipher Figs A11 and A12 in particular.

We have now modified the text to add the definitions of layer boundaries when needed, as for instance on P11 line 8: "The shallowest SACW (between 26.46 and 26.85 kg m-3)...", on P11 line 14 "At IW, between 27.40 and 27.922 kg m-3, transect S..." and on P11 line 16 "...salinity increases especially in charts of diss"olved oxygen versus SA up to more than 27.7 kg m-3 ..."

The original data were interpolated through an inverse model, but the discussion needs to make this point clearer. How do Figs A11 and A12 change if you plot the original biogeochemical data rather than the interpolated data?

All the data represented in these two figures are non-interpolated in situ observations. The density field corresponding to these concentrations is estimated with in situ temperature and salinity which corresponding to the analyzed samples.

This information is included on captions of both figures (A11 and A12) and on P10 line 31: "...where the relationships between in situ measurements of SA, O2, NO3 and PO4..."

The methodology used is as follows: S and T are interpolated with DIVA to gain resolution, after considering the temperature provided by XBTs. The density field is built with them. This density field, reconstructed with both observational and interpolated data, is only used in the inverse model to estimate geostrophic velocities and then mass transports.

The biogeochemical variables, which are not included in the model, are interpolated with DIVA to the points where geostrophic velocities and mass transports are estimated with the model to be able to calculate their transports as the product of mass transport and the concentration of each variable interpolated at this point.

One question is why the authors expect the scaling for nutrients and oxygen concentrations (section 2.4), which are strongly affected by biogeochemical processes, to be the same as those for T and S.

The correlation lengths might not be the same, each process has its own scale. Le Traon (1990) studied how physical processes with the wavelengths less than the correlation scales are filtered with the optimal interpolation, similar to DIVA. In this case, the correlation scales selected were determined by the resolution of the observations. So, the physical or biogeochemical processes with smaller scales can not be studied with these interpolations. However, on large scales such as the case in which we are working, the correlation scales in both physical and biogeochemical processes can be considered to be the same.

I think they should elaborate on this. Another point that is missing is discussion of the imbalance in the transport numbers. While this perhaps can be explained for the oxygen and nutrient numbers on the grounds of biological uptake and nutrient regeneration, this is not

the case for the mass transport. How important is the imbalance of 2.6 + - 4.5 Sv given in Table A1?

21.31% is the relative error of the imbalance of 2.6 +/- 4.5 Sv mainly related with the lack of sypnopticity in the sampling. This sentence is included on P12 line 15 as: "In spite of the 21.31% of the relative error associated to the imbalance of 2.6  $\pm$  4.5 Sv (Tab. A1), mainly related with the lack of sypnopticity in the sampling, more than 60% of the mass transport..."

I have some additional, specific points that will doubtless be picked up by other reviewers: 1. In the methodology section, they say that nutrients were analyzed for nitrate, phosphate and silicate. Were separate nitrite analyses made, as there is commonly a nitrite maximum near the pycnocline in upwelling systems? If not, then I presume that their nitrate concentrations are in fact nitrate plus nitrite.

In this campaign, nitrates, nitrites and ammonium are analyzed separately, but in this paper we only represent the results for nitrates (nitrite and ammonium distribution plots are attached).

**2. On p6, Fig. A3 is introduced before Fig. A2.**

**We have changed the order of these two figures**

3. On p.8, in their discussion of Fig. A4 and in the figure caption, it would help if the authors define the T/S characteristics of the frontal zone.

We compare T and S characteristics in the nearest stations at both sides of the CVF taking 150 m as a reference depth. CVF is located between stations 23 and 24 in transect S and between stations 33 and 34 in transect E. Therefore, we compare T and S at 150 m in these pairs of stations of both transects.

**St 23: 36.37g/kg; 16.38°C**

St 24: 35.66g/kg; 14.45°C

St 33: 35.96g/kg; 15.22°C

St 34: 36.27°C; 16.33°C;

|ΔS (23,24)|=0.71g/kg;

|ΔT (23,24)|=1.93°C;

**|ΔS (33,34)|=0.31g/kg;**

|ΔT (33,34)|=1.11°C;

On P8 line 7 where the location of CVFZ is given, it is included: "First, CVFZ was intersected between stations 23 and 24 where there were important differences in water characteristics,  $\Delta$ SA > 0.70 gkg-1 and  $\Delta$ CT > 1.92°C, between the two sides of CVF at 150 m and then between stations 33 and 34 with  $\Delta$ SA > 0.30 gkg-1 and  $\Delta$ CT > 1.10°C at 150 m"

4. In Figs A4-A6, it is almost impossible to see the pink dots that mark the sample positions as a result of the color distribution used. These figures don't really need color, although I know that plotting routines make it very easy to add. Perhaps a reduced color palette would help here.

We think the colors help with the interpretation of plots. We have changed the color of the sampled points to black and they look better.

5. Fig. A7 has me confused. In the discussion of the transport numbers given in Tables A1 and A2, positive and negative numbers refer to transport out of or into the box respectively. What is going on in Fig. A7? Is the notation the same, or do positive numbers refer to northward or eastward transport across each line as is normally the case? This needs to be stated.

Indeed, they are velocity values perpendicular to the transects with the signs according to the geographical criteria (it is positive northward and eastward), it has already been included on figure (A7) caption and on P9 line2: "The absolute velocity field perpendicular to each transect and with sign depending on geographic criteria (positive sign is northward and eastward) was..."

6. P11, line 11: I think this should be 35.25 g/kg, not 36.25.

Yes, we agree and have modified the text.

7. Given that there is generally flow into the box from the east, how much of this is likely to result from upwelling, which occurs throughout the year in this part of the Canary Current region? Central Water generally provides the source for upwelling waters in similar systems.

We think that the upwelling water exported eastward is quite low as compared to what happens in the rest of the water column. This water is mainly exported in the form of the great filament (Gabric et al, 1993) which is located in the shallowest layers and it is practically all year.

However, the upwelling process itself, south of Cape Blanc is seasonal, intensifying in winter when the trade winds reach the area. North of Cape Blanc, the upwelling is an annual process so we think that some upwelling water may enter into our domain through the northeast corner but not from the east, or not, at least, south of Cape Blanc where there is not upwelling in this season. In fact, during the cruise there were not upwelling-favourable winds.

Despite these questions, most of which are minor, I will send the manuscript out for review as it provides an interesting data set from an understudied region.

We really appreciate this decision.

We appreciate the comments and suggestions provided by the reviewer. We have tried to follow them to produce an improved version of the manuscript. A detailed reply is given below.

1) Can the imbalances be justified in terms of the evolving velocity fields? The authors could compare the numerical and field transports for the times each individual was done (four sections, each carried out in about five days). They could also consider a mean numerical field during the entire period (25 days) and compare it with the combined realizations of the individual sections. Finally, they could use the numerical data to see if, at each time, the transports are balanced.

We have proceeded as suggested by the reviewer, trying to evaluate with numerical model outputs if we might have had a problem of synopticity. On the one hand, we have estimated the imbalance for numerical mass transports in a given day selected in the middle of the cruise and found that the imbalance is as low as only 0.2 Sv (Fig. 1). This result indicates that the numerical model is conditioned to balance mass transports in every time step.

On the other hand, we have produced a synthetic cruise, where we extract the model profiles in the same dates and locations of the cruise, hence introducing a time dependency in the dataset. In this second case, the imbalance is ten times larger, 2 Sv. This result indicates that the time dependency has added some noise in the velocity fields, losing some synopticity (Fig. 2). This imbalance is rather similar to that obtained from the inverse model, suggesting that indeed we might have had a problem of synopticity.

The next version of the manuscript will contain a section to discuss with more detail this issue with the synopticity.

---

## Referee Report (RR1)

**Cape Verde Frontal Zone in summer 2017: lateral transports of mass, dissolved oxygen and inorganic nutrients**

Nadia Burgoa1, Francisco Machín1, Ángel Rodríguez-Santana1, Ángeles Marrero-Díaz1, Xosé Antón Álvarez-Salgado2, Bieito Fernández-Castro3, María Dolores Gelado-Caballero4, and Javier Arístegui5

15

**A1**

[referee-annotated manuscript omitted]

---

## Author Response (AR2)

First of all, we would like to thank again the editor and referees for all corrections and suggestions.

In relation to errors that referee 2 considers we take into account twice, we must clarify that we have followed all the steps documented by Ganachaud in "Error Budget of Inverse Box Models: The North Atlantic" (2003). Specifically, section 3.2b explains the baroclinic variability as a model's error independent of the reference level velocity's error, which is the case we have evaluated in our manuscript.

Regarding the referee 1's comments related to the imbalances in SW and CW: we have run the inverse model considering the upwelling of 0.6 Sv from CW to SW suggested by the referee 1's. The output of the model with this forcing has a higher imbalance (shown below).

We have put in SW: -0.6 Sv and in CW: +0.6 Sv. Hence, we consider that the reviewer was interpreting the transports with the opposite meaning as their actual criteria used along the paper (positive outward).

[Figure]

*Figure 1: Inverse model solution considering the upwelling. A water outlet of 0.6 Sv is forced from CW to SW.*

In any case, as the reviewer is rising a main concern related to the imbalance at SW and CW we have performed an additional analysis focused in reducing the imbalance. To do so, we have tested a change in the reference level to 27.82 kg/m3 and the mass transport imbalances in the first and third layers are reduced (Figure 2). Hence, we have updated the manuscript with this new reference level for the inverse model calculations.

[Figure]

*Figure 2: Inverse model solution considering the reference level at 27.82 kg/m3 and without any forcing.*

In addition, the responses to referee 1's small comments and suggestions are included below.

The manuscript has changed with the modifications for the abstract and with the rest of suggestions proposed by this reviewer.

(4) p. 5, l. 5: "to validate the temperature interpolated to the XBT positions and set the signal to noise ratio..." I agree with the signal-to-noise ratio but you do not use interpolations to validate a measurement but the other way round.

(5) p. 5, l. 8: 10%? Is this correct, 10% and even 5% is far too high: 5% of 20C is 1C, which is very large.

(6) p. 5, l. 13-14, please clarify.

These three comments are addressed by changing the paragraph to this other:

"SP, O2, NO3, PO4 and SiO4H4 were optimally interpolated with DIVA at each transect independently. Before carrying out these interpolations, DIVA was applied to the T field suppressing one xbt profile from each transect to validate the method. In fact, the interpolated values had a relative error<3.5% in 75% of cases. This allows to set the signal to noise ratio ($\lambda$) and the horizontal and vertical correlation lengths (Lx and Ly). Hence, the interpolations of the remaining hydrological and biogeochemical variables were carried out with the following parameters: $\lambda = 4$, Lx = 110–135 km and Ly = 50 m. Despite the fact that each variable behaves differently depending on its physical, chemical or biological nature, the correlation scales were considered the same due to the limitation of the sampling resolution. DIVA provided error maps for the gridded fields of each variable which allowed us to check their accuracy and spatial distribution. 75% of the interpolated values of SP and O2 had a relative error ≤ 3.5%. Due to the lower sampling resolution of NO3, PO4 and SiO4H4, their interpolated values had higher errors. Between 70-75% of the interpolated values had a relative error ≤ 5.7%."

(7) p. 6, l. 20: I'm not sure about how the journal handles citations to manuscripts in preparation but I would say this is highly irregular. I suggest simply to state "(not shown)".

We would like to include this citation, so we will contact and ask to the journal about it.

(8) p. 7, l. 2-3: "to avoid any issues related with the temporal evolution of structures, the volume is closed with land instead of with the eastern transect". Please clarify.

We have changed it with: " to avoid any imbalances induced by the temporal evolution of the system, the volume is closed with land instead of with the eastern transect".

(9) p. 9, l. 27: explain this beforehand, in section 2.2.

The text "The climatology produced with GLORYS outputs is used to present the front spatial distribution within the domain" has moved to section 2.2. like this: "Specifically, the climatological salinity was used to present the CVF spatial distribution within the domain".

(10) p. 10, l. 14-15 and elsewhere (including tables): I suggest not mentioning the results for

DW. You are only sampling a tiny fraction of these DWs, what is the sense of providing these values?

They are included because it has been sampled up to there and every piece of data is valuable.

(11) P. 12, l. 12: Luyten et al. (1983) did not document this shadow zone, they provided a theory that explains its existence. You may rather for example cite Kawase and Sarmiento (JGR, 1985).

We have changed it with this new citation.

(12) Fig. A1: I suggest you show the grid every 1 degree.

We have done it.

(13) Fig. A11a: I suggest to separate SW from CW and remove the DW results.

We have separated SW from CW.

(14) Fig. A15: perhaps you are missing an axis for phosphate transports?

These transports are multiplied by 10 (it is indicated in the image caption).

(15) I suggest adding an additional figure like A15 but for the net values, both within the

original domain and within the domain that excludes the upwelling region.

Instead of including this new figure associated with the problems related to the imbalaces suggested by the referee, we have included a new figure that shows the mass transports per transects and per layer (CW only) in both sides of the front, independently.

[Figure]

*Figure 3: Mass transports ($10^9$ kg s$^{-1}$) integrated per transect at north (N, pink line), west (W, green line) and south (S, blue line) of the subtropical (left) and tropical (right) areas separated by the CVF at the three CW layers and considering transports between WOA stations. Black line represents the net transport. Negative/positive values indicate inward/outward transports as in Fig. A11.*